# Toll-like receptor mediated inflammation directs B cells towards protective antiviral extrafollicular responses

Jonathan H. Lam[1,2,3] & Nicole Baumgarth [1,2,3,4] ✉

Extrafollicular plasmablast responses (EFRs) are considered to generate anti-bodies of low affinity that offer little protection from infections. Paradoxically, high avidity antigen-B cell receptor engagement is thought to be the main driver of B cell differentiation, whether in EFRs or slower-developing germinal centers (GCs). Here we show that influenza infection rapidly induces EFRs, generating protective antibodies via Toll-like receptor (TLR)-mediated mechanisms that are both B cell intrinsic and extrinsic. B cell-intrinsic TLR signals support antigen-stimulated B cell survival, clonal expansion, and the differentiation of B cells via induction of IRF4, the master regulator of B cell differentiation, through activation of NF-kB c-Rel. Provision of sustained TLR4 stimulation after immunization shifts the fate of virus-specific B cells towards EFRs instead of GCs, prompting rapid antibody production and improving their protective capacity over antigen/alum administration alone. Thus, inflammatory signals act as B cell fate-determinants for the rapid generation of protective antiviral extrafollicular responses.

Acute respiratory tract infections induce neutralizing antibody responses that are critical for long-lasting protection. Germinal center (GC) responses are considered the most effective in generating protective antibodies, as antigen-specific GC B cells undergo extensive somatic hypermutation, resulting in long-lived antibody-secreting plasma cells (ASCs) that generate high-affinity, strongly neutralizing antibodies. However, after primary influenza virus infection, GCs appear relatively late, maturing mostly after viral contraction, and thus are unlikely to contribute towards virus clearance[1]. Instead, early antibodies are produced from short-lived plasmablasts of the extra-follicular response (EFR), which develop and localize within the medulla and interfollicular regions[2] of the respiratory tract-draining mediastinal lymph nodes (medLN) shortly after infection and before GC formation[3]. Early studies by Gerhard and colleagues demonstrated that influenza inoculations of BALB/c mice resulted in rapid production of early hemagglutinin (HA)-specific, neutralizing IgG antibodies that were protective and in repertoire distinct from those induced later

in the response[4]. This included unmutated IgG from B cells of the prototypic HA-specific C12 idiotype, which were excluded from GCs after intra-nasal (i.n.) influenza infection[3]. EFRs thus appear physiologically distinct from GCs and can generate protective, germline-encoded, antigen-specific ASCs from the restrictive repertoire of inbred mice.

Addressing how these distinct B cell activation outcomes contribute to humoral immunity against acute respiratory tract virus infections, where rapid induction of immunity is a key determinant of survival, is pertinent for our understanding of the pathogenesis of these infections and the role of B cell immunity. While CD8 T cells are credited as most important for the clearance of influenza virus during primary infection, they alone cannot prevent mortality[5] and may collaterally eliminate non-infected antigen-presenting cells[6]. In addition, lack of B cells led to a ~50 fold increase in virus titers by day 10 post-infection[7], demonstrating the importance of early antibody generation against acute respiratory tract infection. This has important implications for

[1]Graduate Group in Immunology, University of California Davis, Davis, USA. [2]Center for Immunology and Infectious Diseases, University of California Davis, Davis, USA. [3]Dept. Pathology, Microbiology and Immunology, University of California Davis, Davis, USA. [4]Present address: W. Harry Feinstone Dept Molecular Microbiology and Immunology, Johns Hopkins Bloomberg School of Public Health, 615 N Wolfe St, E4135, Baltimore, MD 21205, USA. ✉e-mail: nbaumga3@jhmi.edu

vaccine design, as vaccinations during the ongoing COVID-19 pandemic or during seasonal influenza virus infections are likely more effective if they can induce immune protection more quickly, i. e. through EFRs, in addition to providing long-term immunity through induction of GCs. The signals required for EFR induction, however, have not been resolved. Indeed, EFR induction has been considered of little consequence, as these responses are thought of as only short-lived and of low protective capacity.

Yet, groundbreaking studies by the Hengartner group over two decades ago demonstrated that antibody responses to vesicular stomatitis virus showed a surprising lack of changes in virus-specific serum antibody affinities over the course of infection. Instead, they demonstrated that antibodies of relatively high affinity for their cognate antigen were generated both early and late after infection[8,9], suggesting that following a viral infection, both EFR and GC-derived antibodies might generate antibody responses of overall high affinity. These data are also consistent with reports by the Brink lab, who demonstrated using the BCR-transgenic swHEL model that strong BCR-affinity for antigen drove the rapid proliferation and differentiation of hen egg lysozyme (HEL)-specific B cells in EFRs, while lower affinity interactions induced stronger GC responses instead[10]. Generation of EFRs from high-affinity B cells is consistent also with findings that strong BCR-signaling drives upregulation of interferon regulatory factor 4 (IRF4), a critical transcriptional regulator of plasma cell development[11]. Such a model of affinity-based induction of proliferation and differentiation would be consistent with EFRs' potential to generate high affinity antibodies.

However, whether BCR-antigen interactions alone drive B cell fate decisions towards EFRs remains unknown. Furthermore, in contrast to studies indicating that highly functional antibodies emerge from EFRs, other work has shown that EFRs developing in the spleen following *Salmonella thyphimurium* and *Ehrlichia* infections generate large quantities of predominantly non-specific antibodies[12,13], in support of the idea that EFRs are of little protective consequence. Together, these data seem to indicate that additional infection-induced signals shape EFRs. What these signals are, and how they might affect the functionality and protective capacity of EFR-derived antibodies, is unresolved.

Work interrogating pattern recognition receptors (PPR) signaling after immunization has identified numerous effects on B cells and it is well appreciated that certain PAMPS can work as adjuvants to support vaccine responses. For example, RNA of sheep red blood cells stimulated RNA-sensing PPR mitochondrial antiviral signaling protein (MAVS) and TLR3[14], which supported more robust B cell responses. Also, mice immunized with nanoparticles containing the TLR4 ligand 4'-monophosphoryl lipid A induced a more robust, antigen-specific ASC response compared to mice given antigen alone, while the combination of TLR4 and TLR7 agonists was reported to fate B cells towards early memory and germinal center responses, resulting in persistent antibody responses from bone marrow long-lived plasma cells, rather than rapid EFRs[15]. B cell-intrinsic MyD88 signaling was also shown to increase proliferation and differentiation of plasma cells and induced expansion of Bcl6+ germinal center B cells to virus-like particles[16]. In addition, in Friend virus infection and after infection with influenza virus, B cell-intrinsic expression of TLR7 (but not TLR3) was shown to be required for germinal center formation[17,18]. In contrast, stimulation with the TLR9 ligand CpG antagonized B cell antigen uptake and processing, resulting in disruption of affinity maturation and a reduction in early-formed, antigen-specific plasma cells in the spleen, along with a reduction in long-term, antigen-specific serum IgG avidity[19]. Observations of TLR integration with canonically distinct B cell activation pathways may play a role in the reported effects of TLR agonists on antibody responses, as TLR4 was shown to integrate with BCR signaling via the phosphorylation of syk[20], while the TLR adaptor MyD88 was shown to be critical for signaling via the B cell survival receptor TACI[21]. Collectively, existing evidence suggests that TLR and/or MyD88-mediated signaling affects B

cell responses, but how these signals integrate to regulate B cell responses remains incompletely resolved.

Here we demonstrate that inflammatory signals induced by influenza virus infection, but not immunization with virus particles in alum, triggers the rapid generation of protective antibody responses via formation of EFRs in a TLR signaling-dependent manner. TLR-signaling fated B cells towards the EFR/plasma cell state after infection through the strong induction of IRF4 via activation of NFkB c-Rel. Similarly, sustained co-administration of LPS with virus/alum immunization rescued EFR induction after vaccination and improved antibody-mediated protection against lethal influenza challenge.

## Results

### The extrafollicular B cell response generates antigen-specific antibodies after intranasal influenza infection but not after peripheral immunization

Influenza-specific ASCs are found predominantly in the medLN within 7 days post primary infection (dpi). They are not found in the lungs until 14 dpi[3], well after virus clearance. This indicated that medLN EFRs are the main source of the early antigen-specific antibody response and were thus investigated. At 5 dpi, B cells were predominantly naïve, CD19+/CD45R+ and CD38+/CD24+ (Fig. 1a, left). By 7 dpi, activated pre-GC/GC-like (GC) B cells emerged, identifiable as CD45Rhi/CD19hi and CD24hi/CD38med, along with early-formed, plasmablasts of the EFR (EF PBs), which were identified as CD19lo/CD45Rlo CD24+ CD38lo(Fig. 1a, right). GC precursors also expressed the GC marker GL7 and were high for interferon regulatory factor 8 (IRF8), a transcription factor associated with GC polarization[22] (Fig. 1b, left), while EF PBs were IRF4hi, which is associated with an ASC fate[11,22], with many also expressing CD138 (Fig. 1b, left), a canonical marker of ASCs. EF PBs and GC B cells both had lost surface IgD and most had lost IgM by 7 dpi (Fig. 1b, right), indicating a high level of class-switching. While B cell frequencies in the medLN remained relatively constant throughout the time course (Fig. 1c), drastic changes in EF and GC compartments took place. Relatively few GC B cells were found until after 9 dpi (Fig. 1d), while EF PBs were seen as early as 5 dpi, peaking between 7 and 10 dpi and contracting by 14 dpi (Fig. 1e).

Only EF PBs, purified by flow cytometry, secreted pathogen-specific antibodies at 7 dpi, detected as influenza-bound total Ig and IgG2c by ELISPOT on cells (Fig. 2a), demonstrating that EF PBs contain the only functional, influenza-specific ASCs in the medLN at this timepoint. In addition, use of two distinct fluorophore-labeled, recombinant hemagglutinin (HA) of A/PR8 identified HA-specific (HA) B cells (Fig. 2b) and their preferred participation in EF over GC B cell responses (Fig. 2c–e), with HA-bound B cells comprising as much as 15% of the EFR compartment at the early time points. The independence of EFR formation from GCs during influenza infection, suggested previously[23], was confirmed with the presence of EF B cells in infected Mb-1-Cre Bcl6 f/f mice that are unable to form GCs (Suppl. Fig. 1). Thus, EFRs are responsible for the earliest antigen-specific antibody response to influenza infection and are independent of GCs.

A distinct B cell response was seen after subcutaneous (s.c.) immunization with influenza virions in alum adjuvant. Compared to infection, immunizations yielded an enrichment of GC B cells over minimally induced EFRs in the draining LN at 3, 7 and 10 dpi (Fig. 3a). GC B cell numbers, but not EFRs, increased in an antigen-dose dependent manner (Fig. 3b). Furthermore, while there was expansion of HA B cells after immunization, they predominantly exhibited a GC but not EFR phenotype (Fig. 3c–e). We conclude that infection-induced signals are required for the generation of robust EFRs.

### Global ablation of MyD88/TRIF signaling drastically alters EFR kinetics to influenza

To identify the influenza infection-induced signals that support EFRs, we first considered inflammatory cytokines that were previously identified

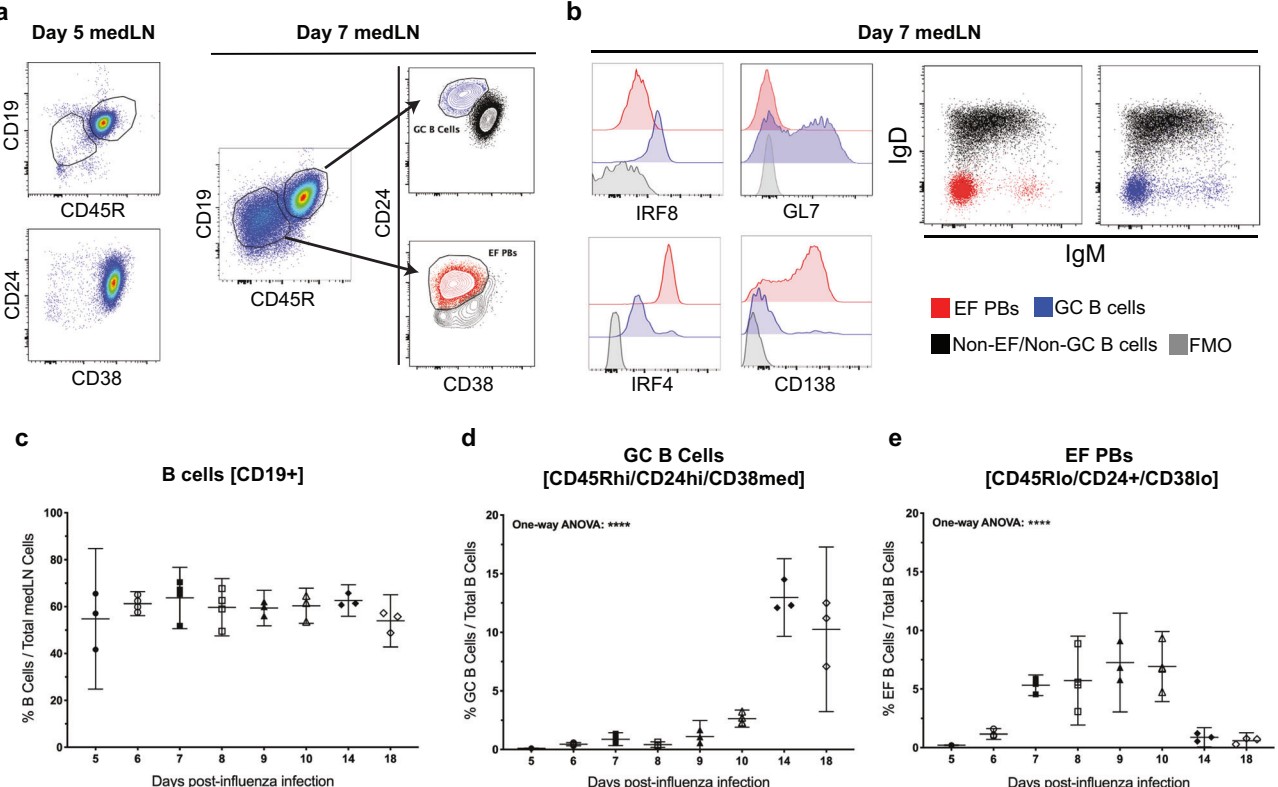

**Fig. 1 | Primary influenza infection induces strong early EFRs prior to GC formation.** Shown are flow cytometric analyses of mediastinal lymph nodes (medLN) from C57BL/6 mice infected with influenza A/PR8 intra-nasally (i.n.) at seven days post-infection (dpi). **a** Identification of extrafollicular plasmablasts (EF PBs) by gating first on CD19lo/CD45Rlo then CD24+/CD38- and pre-GC/GC B cells on CD19+/CD45R+ then CD24hi/CD38lo. Presence of CD19lo/CD45Rlo and CD24+/CD38- populations at 5 dpi (left) and 7 dpi (right). **b** IRF8 and GL7 expression shown to confirm GC identity and IRF4 and CD138 expression shown to confirm EF PB identity (left), along with IgM and IgD expression of B cell subsets (right). **c–e** C57BL/6 mice (*n* = 3–4) were infected and medLN were collected on the days specified, measuring B cell frequencies of total cells (**c**), pre-GC/GC frequency of B cells (**d**), and EF frequency of B cells (**e**). Data in (**c–e**) represent mean ± 95% confidence interval (CI) of two independent experiments. Statistical significance determined by one-way ANOVA. ****$p < 0.0001$. Source data are provided as a Source Data file.

as contributing towards B cell differentiation and ASC maintenance, as well as S100A9, a damage-associated molecular pattern protein produced by stressed and dying cells and released during influenza infection[24]. Among the cytokines tested, IL-1, Type I interferons (IFN), IL-6, and TNFα are induced early after influenza infection[25–27] and support ASCs[28–30]. IL-12 and the effector cytokine it supports, IFNγ, which is produced by T cells, NK cells, and ILC1, are known to support ASC maintenance[23,31]. Mice deficient in each of these soluble cytokines or their receptors showed EFRs similar to their wild type (WT) controls at 7 dpi (Suppl. Fig. 2a), while mice lacking TNFα signaling showed significantly increased GC responses. B cells are also importantly affected through innate signals received via Toll-like receptors (TLRs). Influenza pathogen-associated molecular patterns (PAMPs) activate endosomal TLR3[32] and TLR7[33], and TLR4 has a role in infection-mediated pathology[34]. However, mice lacking either TLR3, TLR4, or TLR7 showed similar total EF PBs frequencies, and CD138 + EF PBs as their WT controls (Suppl. Fig. 2b). In fact, there was a slight but significant increase in EFRs of TLR7 KOs. Thus, individual cytokines or innate signaling receptors appeared either unnecessary or redundant for EFR development.

The potential for redundancy of inflammatory signals contributing to the regulation of EFRs was addressed with mice double-deficient for both TLR adaptors, TRIF[35] and MyD88 (DKO), which also transduces IL-1 and IL-18 signaling[36]. Indeed, DKO mice showed strongly reduced EFRs at 7 dpi (Fig. 4a). In contrast, TRIF single knockouts had nominal EFRs, while MyD88 single knockouts EFRs were reduced on average, but not significantly (Fig. 4a).

Importantly, serum from WT mice at 10 but not 0 dpi provided robust protection against a lethal influenza virus challenge after passive transfer, a time point at which most, if not all, antibodies are EFR-derived, while the functional protective capacity of serum antibodies collected from DKO mice at 10 dpi was significantly reduced (Fig. 4b, c). Surprisingly, infection of another TLR-null model, through deletion of genes for TLR2[37], TLR4[38] and a missense mutation of Unc93b[39] (TKO), showed EFRs similar to WT controls (Fig. 4a) along with nominal passive protective capacity (Fig. 4d), despite slight reductions in CD138 + EF PBs at 7 dpi (Fig. 4a).

To distinguish potential B cell extrinsic from intrinsic effects of TLR signaling on EFR induction, mixed bone marrow irradiation chimeras (BMC), in which only B cells lacked either MyD88 plus TRIF (DKO BMC) or all TLRs (TKO BMC), were infected with influenza and analyzed at 7 dpi (Fig. 5a). Both the DKO and the TKO BMCs showed reduced EF and GC responses compared to WT chimera controls (Fig. 5b). The data suggested similar B cell intrinsic roles for MyD88/TRIF and upstream TLR signaling in regulating B cell responses overall. However, virus titers at 10 dpi were no different between control and TLR-null BMCs (Suppl. Fig. 3a), while global DKO and TKO mice demonstrated a lack of virus control relative to wild type (Suppl. Fig. 3b). These higher virus titers correlated with significantly larger EFRs in both types of TLR-null mice compared to controls at a late timepoint (Suppl. Fig. 3c). Thus, B cell-intrinsic TLR signaling affects early EFR formation, but lack of virus control, due to a global lack of TLRs, correlates with enlarged but severely delayed EFRs.

### B cell intrinsic TLRs support B cell proliferation and survival
To assess the direct effects of TLR signaling on B cell dynamics, negatively enriched naïve, follicular B cells were cultured with anti-IgM

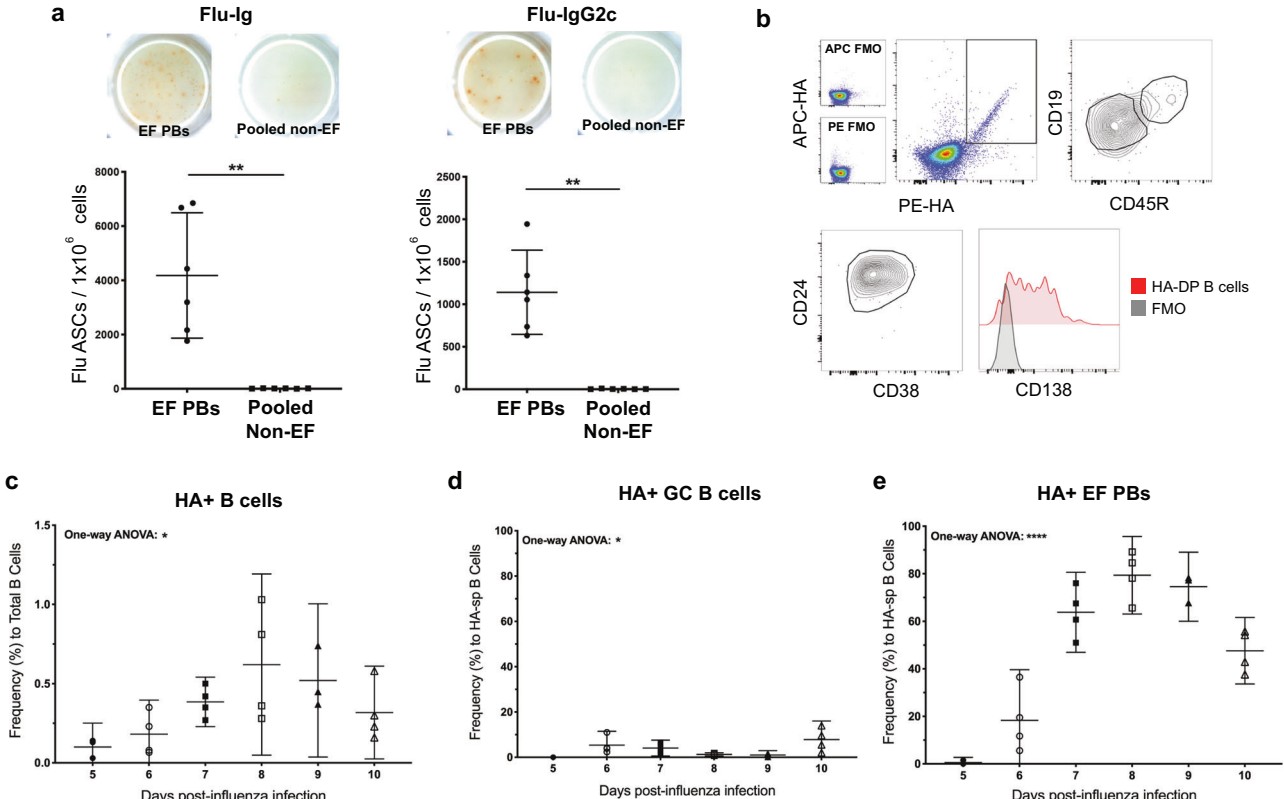

**Fig. 2 | EFRs generate influenza-specific antibody-secreting cells. a** Influenza-specific ELISPOTS of sorted EF PBs and pooled non-EF cells from influenza-infected C57BL/6 mice ($n = 6$) for total Ig (left) and IgG2c (right). **b** Flow plots of HA-specific B cells identified using double HA-tetramer staining with subsequent phenotyping by additional markers as outlined in Fig. 1a. **c**–**e** Time course of HA-specific B cell subsets during influenza infection in C57BL/6 mice ($n = 3$–4), measuring frequency of HA-specific clones (**c**), HA-specific pre-GC/GC clones (**d**), and HA-specific EF PBs (**e**). Data in (**a**, **c**–**e**) represent mean ± 95% CI of two independent experiments. Statistical significance determined by two-tailed Student's $t$-test with Welch's correction or one-way ANOVA. *$p < 0.05$, **$p < 0.01$, ****$p < 0.0001$. Source data are provided as a Source Data file.

(Fab)$_2$ and LPS, BCR and TLR agonists, respectively. Anti-IgM plus LPS co-treatment only slightly enhanced cell viability compared to LPS alone (Suppl. Fig. 4a), but strongly supported B cell proliferation, as indicated by increased Ki67 expression, compared to either treatment alone (Suppl. Fig. 4b). Co-stimulation also had a sizable additive effect over either stimulus alone on IRF4 induction and a more modest induction of IRF8, both critical transcriptional regulators of B cell fate (Suppl. Fig. 4c, d). Taken together, intrinsic TLR stimulation enhances BCR-mediated activation and proliferation.

Canonical TLR signaling is known to integrate with the BCR[20,40] and with TNF superfamily receptors[21], suggesting that TLR signaling-deficient B cells are altered not only in their response to TLR agonists, but also to signals induced via the BCR, or via co-stimulation through CD40 or BAFFR. Indeed, stimulation of naïve, follicular DKO and TKO B cells pulsed with anti-IgM(Fab)$_2$ for three hours, followed by incubation with CD40L and BAFF for 48 h (Fig. 5c) showed reduced viability (Fig. 5d, top) and a near inability to enter the cell cycle, as measured by Ki67 staining (Fig. 5d, bottom), compared to WT controls. MyD88 and TRIF single KO B cells showed reductions in survival (Suppl. Fig. 5a) and proliferation (Suppl. Fig. 5b) similar to each other with frequencies approximately half between those of WT and DKO B cells, indicating that TRIF, along with MyD88, support BCR-mediated activation signals in a non-redundant, additive manner. Similar results were obtained with BCR-stimulation alone (Suppl. Fig. 5c, d), demonstrating participation of the TLR signaling axis in antigen-mediated activation. Consistent with these data, analysis of non-EF/GC B cells from influenza infected DKO and TKO B cell chimeras revealed significantly reduced expression of Ki67 ex vivo, compared to controls at 5 dpi (Fig. 5e), just before rapid plasmablast expansion begins (Fig. 1e). Thus, lack of integrated TLR signaling significantly reduced B cell survival and cell cycle entry consistent with earlier reports[20].

## Lack of functional TLR signaling leads to abnormal BCR complex dynamics and transcriptional control

BCR-mediated calcium flux, an immediate readout of BCR cross-linking, was overall comparable between WT, DKO or TKO B cells (Suppl. Fig. 6a), indicating that TLR-null B cells were not merely defective overall. Effector protein phosphorylation downstream of the BCR showed minor pre-treatment differences (Suppl. Fig. 6b). However, TLR-null B cells generally saw increases in BCR-mediated pathway activation, with the pro-inflammatory NFkB1[41] and the pro-growth regulator mTOR[42] having increased phosphorylation in B cells lacking MyD88 at all concentrations of anti-IgM treatment after 30 min (Suppl. Fig. 6c, d). In addition, Syk phosphorylation, a major activation node for several BCR-mediated signaling pathways[43], was increased in all TLR-signaling deficient B cells at the highest concentration of BCR stimulation (Suppl. Fig. 6e), while there was little difference in the mitogenic pathway MAPK p38[44] between strains (Suppl. Fig. 6f). Together, these data indicate that TLR signaling defects have no detrimental impact on the induction of many important, IgM-BCR mediated signal transduction pathways.

IRF4 is upregulated proportionally to BCR signaling strength[11]. Consistent with that, ex vivo analysis of EF plasmablasts at 7 dpi showed their distinct higher expression of IRF4 and intermediate expression of IRF8 compared to non-EFR B cells (Fig. 6a, left). Non-differentiated B cells from DKO and TKO chimeras expressed significantly less IRF4 and IRF8 than WT at 5 dpi (Fig. 6a, right), indicating defects in IRF4 upregulation just as nascent EFRs begin to form. Ex vivo

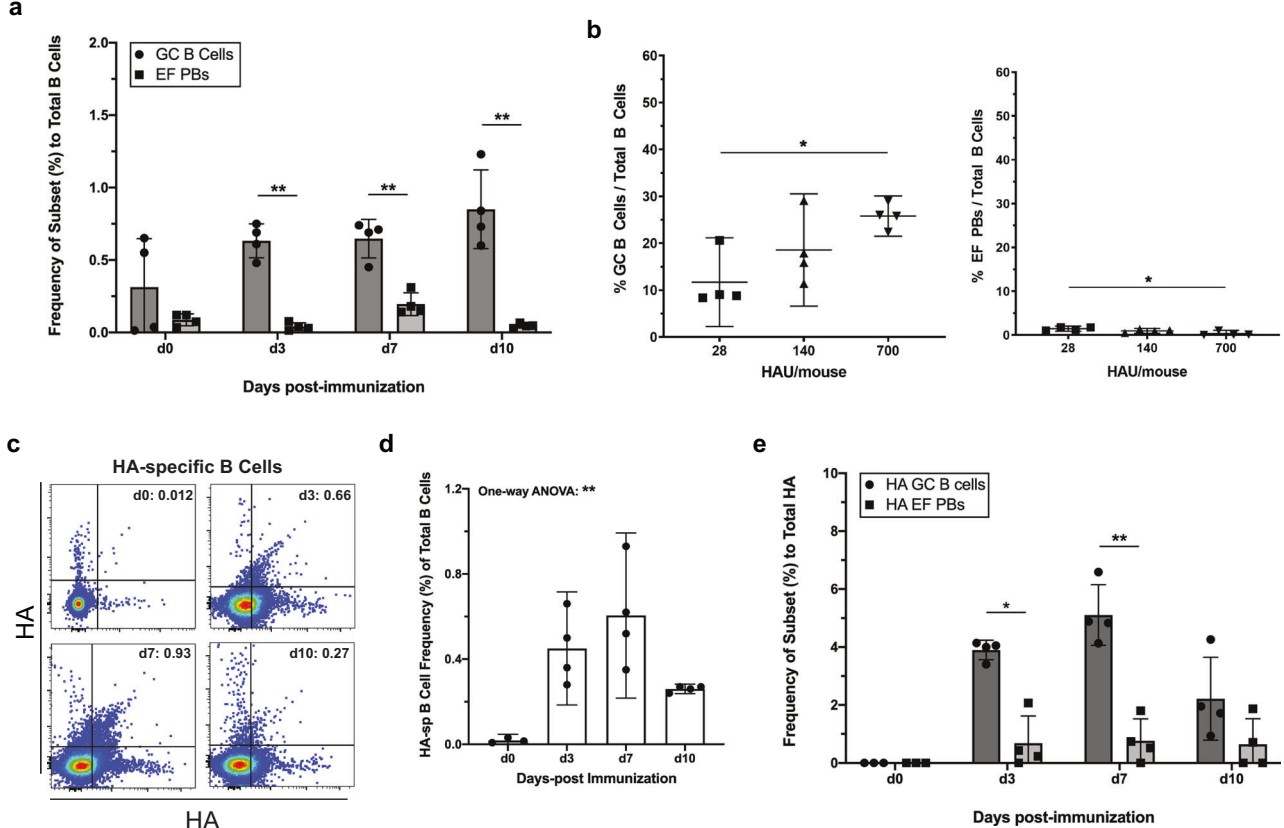

**Fig. 3 | Subcutaneous immunization with influenza and alum does not elicit EFRs. a–e** C57BL/6 mice ($n = 4$) were immunized s.c. with $1 \times 10^7$ PFU influenza A/PR8 in alum and inguinal LNs were analyzed on days indicated. **a** Comparison of EF PB and GC B cell frequencies at multiple timepoints post-immunization. **b** Mice ($n = 4$) were immunized s.c. with indicated HAU sucrose-gradient purified influenza A/PR8 virion emulsified in Complete Freund's adjuvant and analyzed at 7 dpi. GC B cell (left) and EF PB (right) frequency were compared at each antigen concentration. **c** Representative flow plots gating for HA-specific B cells at specified timepoints after immunization in (**a**). **d** Quantification of HA-specific B cells at each timepoint post-immunization. **e** Frequencies of HA-specific B cells that were either GC or EF compared by timepoint. Data in (**a**, **b**, **d**, **e**) represent mean ± 95% CI of two independent experiments. Statistical significance determined by one-way ANOVA and two-tailed Student's *t*-test with Welch's correction. *$p < 0.05$. **$p < 0.01$. Source data are provided as a Source Data file.

baseline levels of IRF4 in naïve B cells were similar between all strains (Fig. 6b, left)., In vitro, IgM-BCR stimulation increased IRF4 and IRF8 expression in B cells from WT mice in an anti-IgM dose-dependent manner in the presence of CD40L and BAFF (Fig. 6b, right, Fig. 6c). Strikingly, B cells from DKO and TKO mice failed to upregulate IRF4 under these conditions (Fig. 6b, right, Fig. 6c), while IRF8 expression remained more similar in all strains (Fig. 6b, right, Fig. 6d). The data thus indicate defective BCR-mediated IRF4 induction in the absence of TLRs.

NF-kB c-Rel is known to promote IRF4 expression upon nuclear localization and is downstream of both BCR and TLR4[45]. Strong, BCR dose-dependent stimulation-induced reductions in cytoplasmic c-Rel, inferring translocation of c-Rel to the nucleus, were seen in WT but much less so in TLR-signaling deficient B cells by flow cytometry as early as 30 min post stimulation (Fig. 6e, Suppl. Fig. 7a). Consistent with that result, DKO and TKO B cells showed significant reductions in nuclear accumulation of c-Rel 1 h after anti-IgM and LPS stimulation, as assessed by ELISA on isolated nuclear-fractions (Suppl. Fig. 7b), but normalized after 2 h (Suppl. Fig. 7c), which was concomitant with significant increases in total c-Rel expression (Suppl. Fig. 7d). However, this delayed normalization in BCR-induced c-Rel expression was short-lived, as sustained c-Rel expression, which is associated with initialization of the B cell differentiation program[46], remained drastically lower in B cells lacking TLR-signaling than in WT B cells 48 h after anti-IgM pulse (Fig. 6f). Thus, even in the absence of deliberate addition of a TLR agonist, B cells require the presence of TLRs for proper activation of the c-Rel circuitry and for the long-term maintenance of c-Rel expression in response to antigen-mediated stimulation.

## Reconstitution of EFRs during influenza immunization through LPS adjuvant

Since both B cell-intrinsic and -extrinsic TLR signals influenced EFR magnitude and kinetics, we tested whether LPS, a TLR4 agonist that initiates both MyD88 and TRIF signaling, could overcome the lack of EFRs induction after s.c. immunization with influenza virions in alum (Fig. 3). Indeed, C57BL/6 mice inoculated with influenza in alum plus LPS, and provided with repeated LPS boosts thereafter (Ag+LPS; Fig. 7a), showed increased total B cells, GC B cells, and EF PBs compared to mice receiving influenza in alum alone (Ag Only) (Fig. 7b). Importantly, the number of HA-binding B cells were twice as high than in mice receiving antigen/alum alone (Fig. 7c), with several-fold increases of HA B cells in the EFR but not GC compartment (Fig. 7c).

The data thus indicate that TLR activation not only increased the expansion of antigen-specific B cells but preferentially shunted them towards an EFR fate. HA B cells from Ag+LPS mice were mostly positive for Ki67/CD138 and were IRF4hi IRF8int., similar to EF PBs from influenza-infected mice (Fig. 7d, e). This level of EFR polarization was not seen in Ag Only mice (Fig. 7d, e). Thus, sustained TLR-mediated inflammation in the presence of antigen leads to greater expansion of antigen-specific B cells and polarizes them towards the EFR fate.

Recent reports suggest that increased antigen valency[47] and antigen availability[48] bias B cells towards a plasmablast fate. Given the

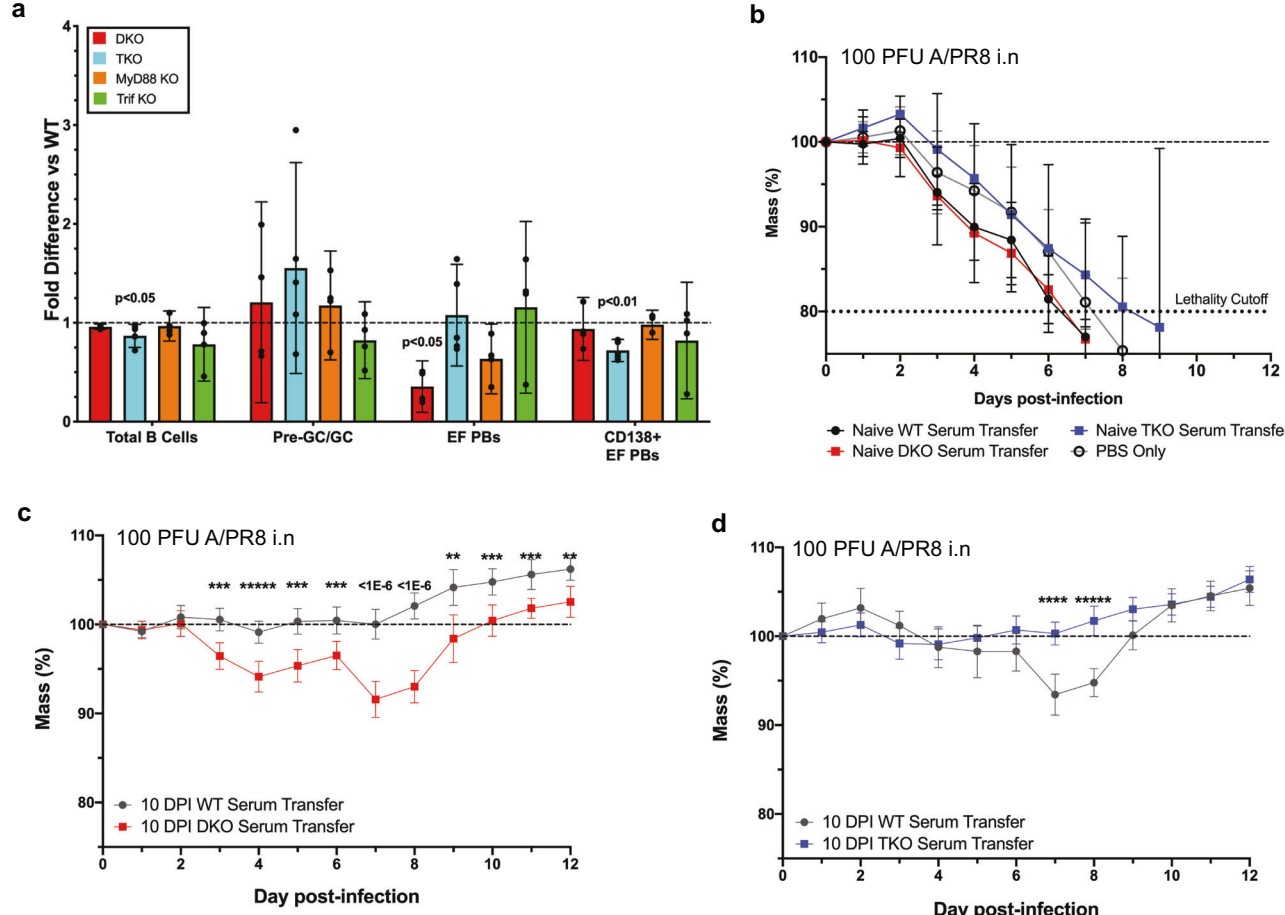

**Fig. 4 | Optimal EFR kinetics and protective antibodies require MyD88 and TRIF.** Knockout and WT mice ($n = 4-5$) were infected with 10 PFU A/PR8 and medLNs were collected at 7 days post-infection (dpi). **a** Fold-difference of B cell subsets in TLR-deficient versus WT mice at 7 dpi. **b–d** Serum was transferred to C57BL/6 mice prior to infection with a lethal dose (100 PFU) of influenza A/PR8 the next day from naïve WT, DKO, and TKO mice ($n = 4$) (**b**) and influenza-infected, age/sex matched- WT and MyD88/TRIF-deficient (DKO) mice ($n = 10$) (**c**) or WT and

TLR2/4/unc93b-deficient (TKO) mice ($n = 10$) (**d**) from 10 dpi. Shown is percent change in weight over the course of infection. Data in (**a–d**) represent mean ± 95% CI of two independent experiments. Statistical significance determined by two-way ANOVA and two-tailed Student's *t*-test with Welch's correction. **$p < 0.01$, ***$p < 0.001$, ****$p < 0.0001$, *****$p < 0.00001$ or indicated in subfigures. Source data are provided as a Source Data file.

above results, we asked how B cell fate dynamics and EFR-derived antibody functionality is affected by repeated antigen exposure with or without TLR agonist provision. For that, all mice were primed with influenza and LPS to ensure equivalent initiation of LN activation[49], followed by two additional boosts with antigen alone (Ag Boosted), or antigen plus LPS (Ag+LPS Boosted) or LPS alone (LPS Boosted) as a control (Fig. 8a). Both Ag Boosted and Ag+LPS Boosted mice had similar frequencies of HA B cells in the draining LN (Fig. 8b), and roughly similar frequencies of Ki67+ cells (Fig. 8c). However, HA B cells from Ag Boosted mice significantly polarized towards a GC fate (Fig. 8d), while HA B cells from Ag+LPS Boosted mice polarized significantly towards EFRs (Fig. 8e), indicating that despite repeated antigen inoculations, continued TLR stimulation was required for B cell development towards an EFR fate. Ag+LPS Boosted mice did not see any decreases in GCs at 14 dpi compared to Ag Boosted (Suppl. Fig. 8), suggesting that TLR signaling was not detrimental to the latent GC response, despite early polarization towards EFRs.

Ag + LPS Boosted mice had the highest levels of serum anti-influenza antibodies (Fig. 8f), demonstrating that increased EFRs correlated with enhanced antigen-specific antibody responses compared to a GC-biased response at 10 days post-prime. To determine whether the increase in IgG levels correlated with increased serum passive protective capacity, pooled serum from each boosted group was transferred to naive animals, which were subsequently challenged with

a lethal dose of influenza. Mice receiving Ag+LPS Boosted serum showed no mortality, in contrast to mice receiving Ag Boosted or LPS Boosted serum (Fig. 8g), with results from the latter cohort making cytokine-related protection from LPS unlikely. Moreover, mice that received serum from Ag+LPS Boosted mice lost significantly less weight overall than mice receiving serum from Ag Boosted animals (Fig. 8h). Together, these data demonstrate that sustained TLR-mediated inflammation polarizes antigen-specific B cells towards the EFR, leading to faster and stronger increases in protective, antigen-specific serum antibodies.

## Discussion

These studies demonstrate that TLR-mediated inflammatory signals direct antigen-specific B cells towards the formation of ASCs through EFRs. Importantly, EFR-derived antibodies, induced after both influenza infection and following LPS-boosted immunization, were functionally protective. Thus, EFRs triggered and supported by inflammatory stimuli can provide a protective antibody response at a fraction of the time required for GCs, forming actively secreting, hemagglutinin-specific plasmablasts during the first 6–10 days of influenza infection prior to the formation of GCs, a kinetic that correlates with virus clearance.

EFR development seems to be driven by specificities already present in the repertoire at the time of infection, including in a naïve repertoire[8–10]. In support, high affinity interactions between the BCR

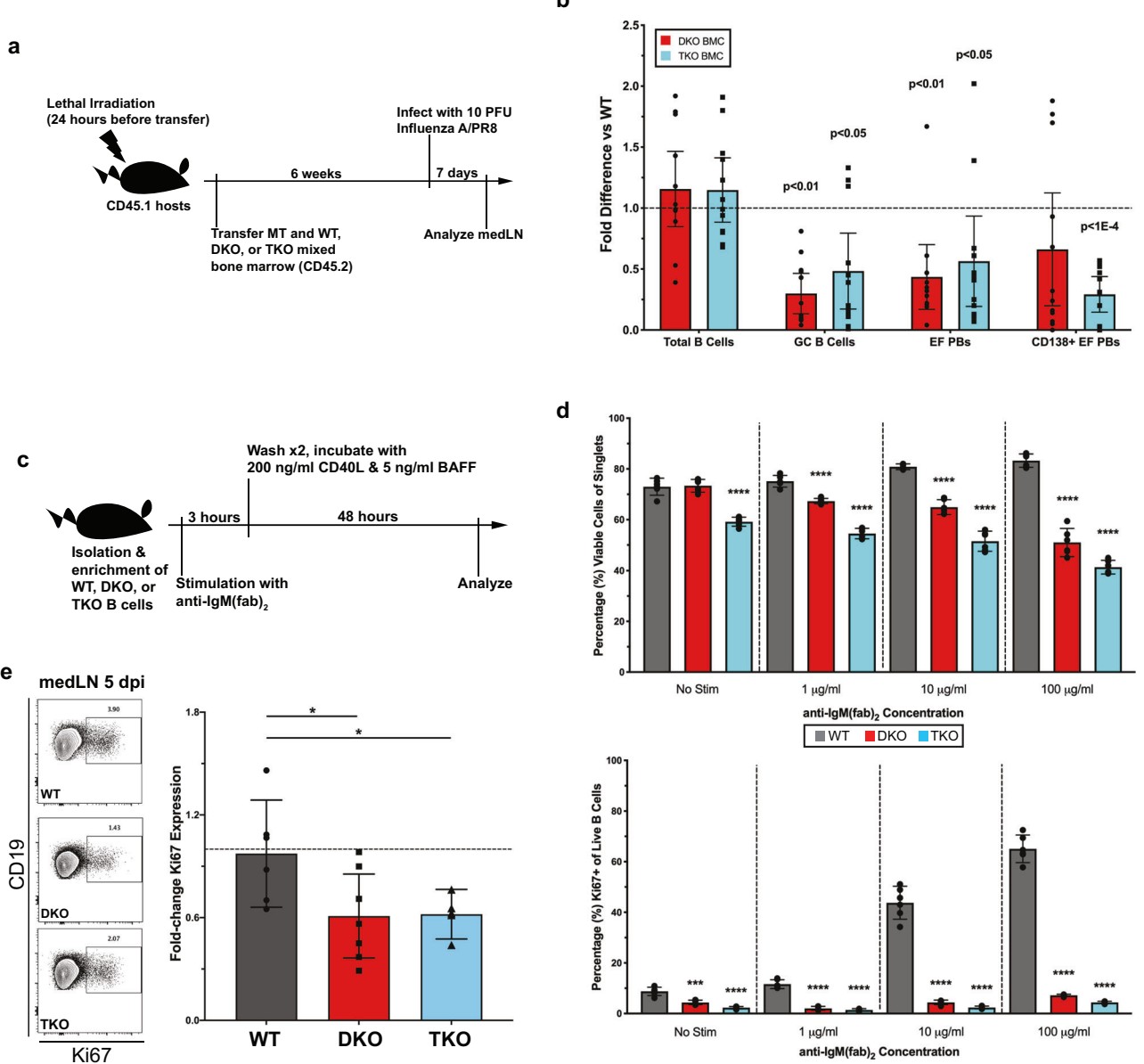

**Fig. 5 | BCR-mediated survival and proliferation are defective in the absence of TLR signaling. a** Mixed bone-marrow chimeras (BMC) (n = 12) established with irradiated CD45.1 C57BL/6 host mice reconstituted with μMT donor BM and BM from either DKO or TKO, then infected with 10 PFU A/PR8 6 weeks later. **b** Quantification of DKO and TKO BMC compared to WT BMC controls of B cell subsets at 7 dpi. **c** Negatively enriched (>98% purity), pooled splenic and LN B cells from WT, DKO, or TKO mice (n = 2–3) were pulsed with graded levels of anti-IgM for 3 h, then stimulated with CD40L and BAFF for 48 h. **d** Quantification of cell viability (top) and cell proliferation (bottom). **e** Ki67+ non-EF/GC B cells in chimeras (n = 5–7) from 5 dpi. Data in (**b**, **d**, **e**) represent mean ± 95% CI of two (**d**, **e**) or three (**b**) independent experiments. Data in (**d**) contain n = 6 total replicates per group. Statistical significance determined by one-way ANOVA and two-tailed Student's t-test with Welch's correction. *p < 0.05, ***p < 0.001, ****p < 0.0001 or indicated in subfigures. Source data are provided as a Source Data file.

and its cognate antigen can drive a B cell effector fate, while lower affinity interactions confers a predispositon for the GC[10]. However, the presence of high avidity B cells alone unlikely explains B cell fate decisions, as we show here that GC formation dominated early B cell responses to influenza immunization, while EFR dominated responses after influenza infection in the same inbred mice. If antigen-BCR affinity alone were to drive polarization towards an ASC fate, then the presence of antigen alone, assuming optimal delivery, stability, etc., should have resulted in an appreciable expansion of the same high affinity clones into the EFR that we saw after infection, yet that is not what was observed.

Together, the data presented here demonstrate the need for infection-induced inflammation as a critical positive regulator of EFR

development. Inflammation affected EFR induction in a B cell-intrinsic manner, as functional Toll-like receptor (TLR) signaling axes, either through MyD88/TRIF or TLR2/4/Unc93b, induced optimal activation of the NF-kB c-Rel:IRF4 pathway (Suppl. Fig 9, top). In addition, it also seemed to act in an extrinsic manner, where TLR-mediated inflammation drives expansion of antigen-specific B cells into the EFR over the GC (Suppl. Fig 9, bottom), perhaps through alterations of the LN stromal compartment[49]. Differences in virus clearance between global TLR-null mice and B cell-specific, TLR-null chimeras underscore the extrinsic and intrinsic contributions of TLR signaling, with B cell-specific TLR-null chimeras having replete TLR signaling in non-B cells that allowed for optimal innate and T cell responses, presumably explaining their reduced viral loads compared to the total TKO mice. It

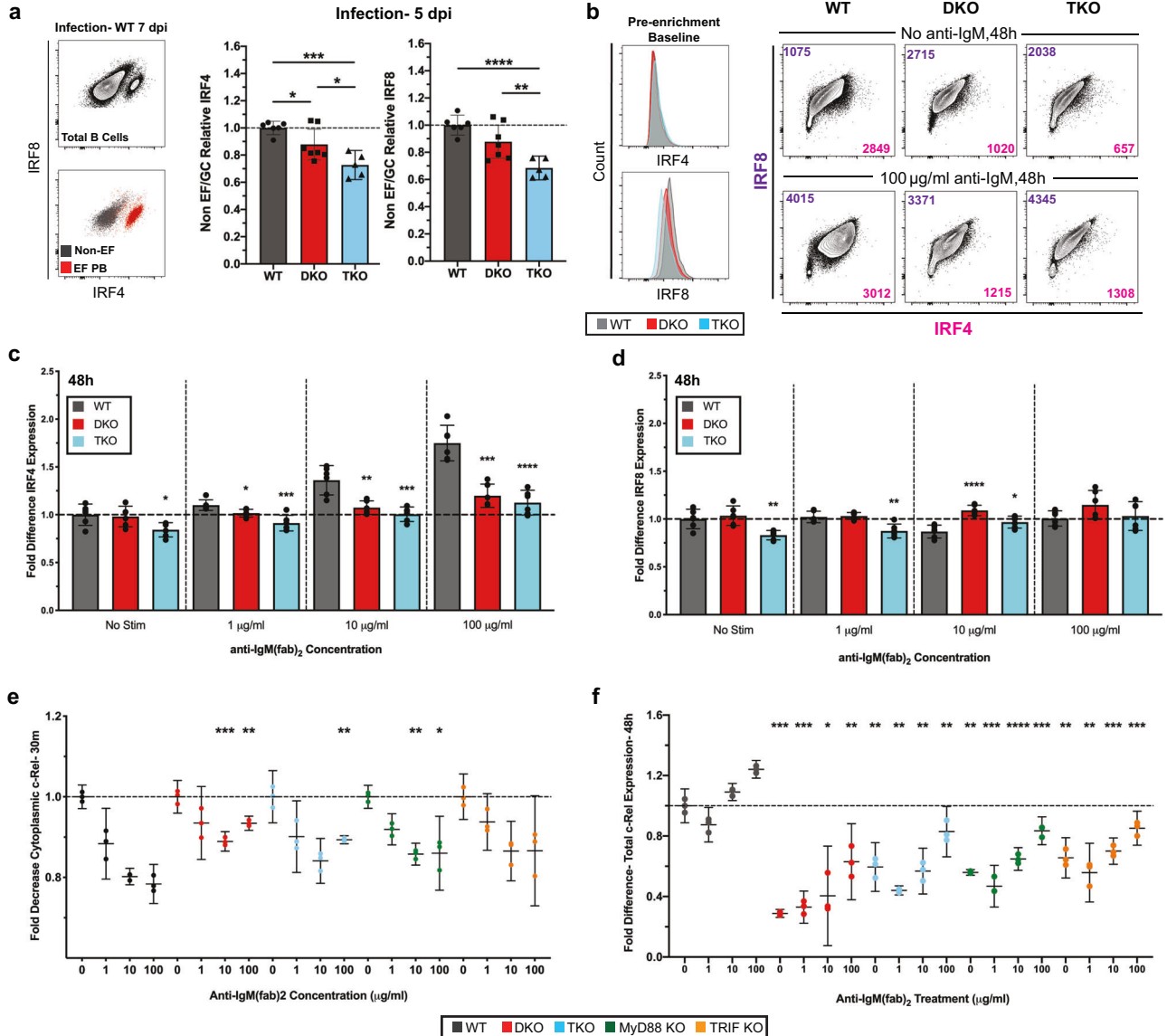

**Fig. 6 | Lack of functional TLR signaling leads to altered BCR complex dynamics and failure to upregulate IRF4.** **a** Representative flow plots showing IRF4 and IRF8 expression in infected mice, highlighting clustering of EF PBs (left). Fold-difference compared to WT controls in IRF4 and IRF8 of non-EF/GC B cells from chimeras ($n = 5–7$) at 5 dpi (right). **b** Pre-enrichment baseline of IRF4 and IRF8 in B cells of each strain (left) and representative IRF4 versus IRF8 flow plots from cells stimulated with indicated anti-IgM concentrations (right). Colored numbers in plots correspond to each like-colored axis. **c**–**d** Fold-difference compared to non-stimulated WT B cells of IRF4 (**c**) and IRF8 expression (**d**) after treatment outlined in Fig. 5c. **e** Fold-difference of knockout versus non-stimulated WT B cells in cytoplasmic c-Rel measured by flow cytometry after 30-minute anti-IgM or LPS treatment ($n = 3$). **f** Fold-difference compared to non-stimulated WT B cells of total c-Rel expression after a 3 h anti-IgM pulse and 48 h culture in complete media only ($n = 3$). Data in (**a**, **c**–**f**) represent mean ± 95% CI of two independent experiments. Data in (**c**, **d**) contain $n = 6$ total replicates per group. Statistical significance determined by one-way ANOVA and two-tailed Student's $t$-test with Welch's correction. *$p < 0.05$, **$p < 0.01$, ***$p < 0.001$, ****$p < 0.0001$. Stars in (**g**, **h**) are Student's $t$-test comparison to respective WT control. Source data are provided as a Source Data file.

is of interest that GC responses were unaffected in global TLR knockout mice but not in the B cell-specific knockouts. While speculative, the data may suggest that a lack of TLR signaling in certain non-B cell populations led to a rescue of GC B cells but not EF PBs. As TLR signaling in DCs leads to increased Th1 polarization[50], total ablation of TLR signaling may have polarized more CD4 T cells towards a Tfh phenotype, compensating for the GC B cell-intrinsic defect in TLR-mediated activation. Further work is required to explore these findings.

TLR stimulation leads to the activation of multiple gene programs, but a defect in NF-kB c-Rel nuclear localization and upregulation after BCR stimulation was specifically observed in DKO and TKO B cells, along with suboptimal survival and the inability to proliferate or

induce IRF4 expression. In addition, the TLR adaptor TRIF was demonstrated here to contribute equally and non-redundantly with MyD88 towards B cell survival and proliferation after anti-IgM treatment. Evidence of enhanced TLR9-MyD88-BCR complexing in activated B cell-like lymphoma cells suggests that TLRs may provide a platform for downstream TLR targets to become activated through the BCR and its effector pathway[51]. Consequently, this would make activation of the integrated TLR pathway a critical source of BCR-mediated IRF4 induction.

The observed defect in IRF4 upregulation in TLR-null B cells is consistent with previous studies demonstrating the dependence of IRF4 induction on c-Rel nuclear translocation after both, TLR4 and BCR activation[45]. Delayed normalization of BCR-mediated c-Rel localization

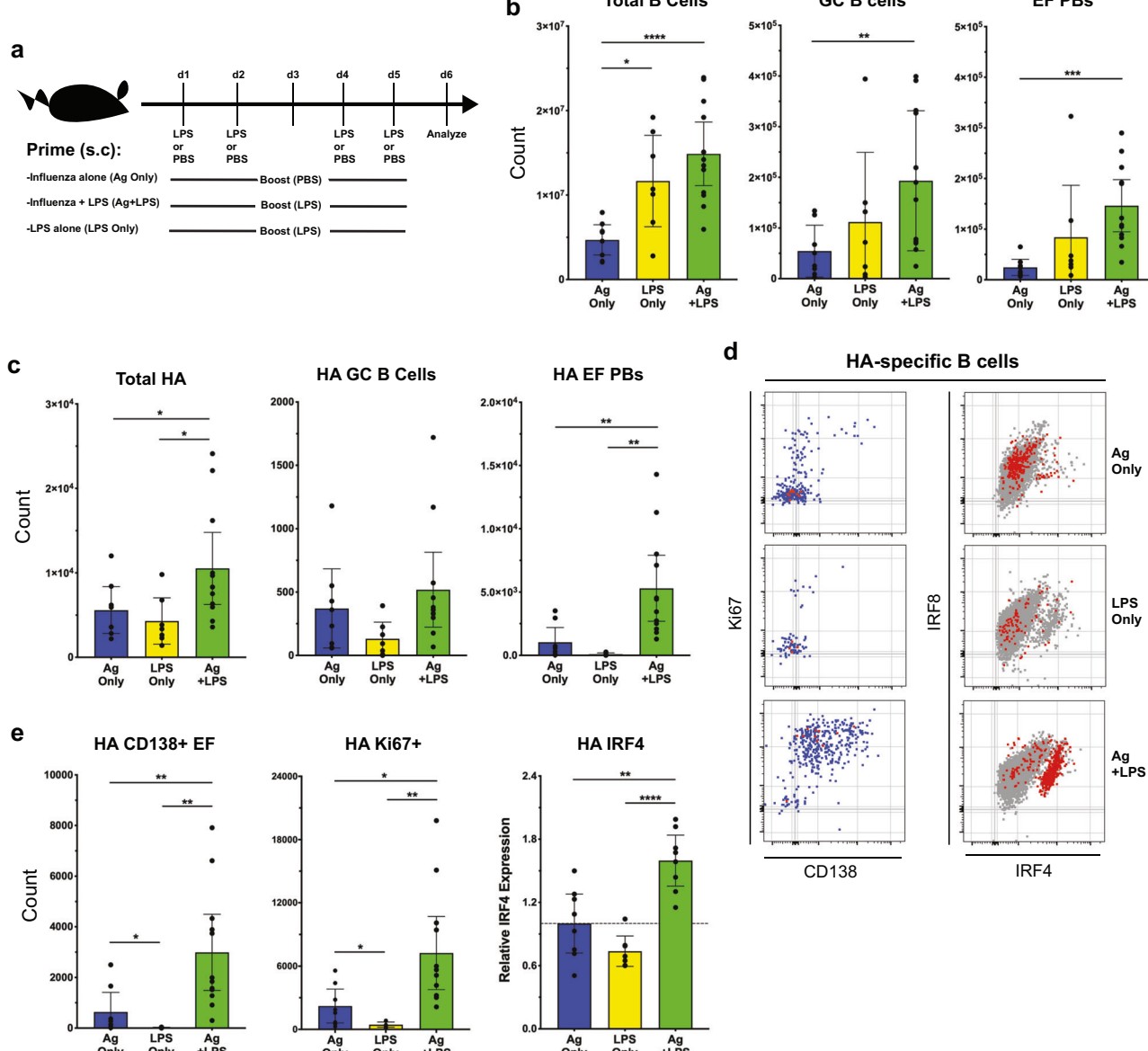

**Fig. 7 | Sustained TLR-mediated inflammation generates strong EFRs in the draining LN after immunization. a** Mice ($n$ = 7–12) were immunized s.c. with or without influenza in alum and with or without LPS, then boosted with either LPS or PBS on days specified, followed by analysis of draining LN. **b** Counts of major B cell subsets. **c** Quantification of HA-specific B cell subsets as in (**b**). **d** Flow plots of HA-specific B cells from each regimen in terms of proliferation and plasma cell differentiation (left) and IRF4 vs IRF8 signature (right, HA-sp. highlighted in red). **e** Quantification of HA-specific EF PBs, proliferation, and relative expression of IRF4. Data in (**a**, **c**–**e**) represent mean ± 95% CI of two (**e**[right]) or three (**b**, **c**, **e** [left, middle]) independent experiments. Statistical significance determined by one-way ANOVA and two-tailed Student's $t$-test with Welch's correction. *$p < 0.05$, **$p < 0.01$, ***$p < 0.001$, ****$p < 0.0001$. Source data are provided as a Source Data file.

in TLR-null B cells did occur two hours after initial stimulation. Given that c-Rel has multiple c-terminal phosphorylation sites[52], TLR components might be required for an optimal phosphorylation signature in addition to release of c-Rel from IkBs. Indeed, it was observed that the regulatory activity of c-Rel carrying a truncated c-terminus was severely altered, despite functional dimerization, nuclear localization, and DNA binding[53]. Therefore, ablation of a functional TLR axis may dictate the nuclear activity of c-Rel, while maintaining localization potential. Further work is needed to determine how TLRs affect phosphorylation of the c-terminal trans-activation domain of c-Rel and how specific gene regulation is altered in their absence. In addition, while total c-Rel levels did increase after 48 h in TLR-null B cells, they were still significantly below the levels observed in the respective WT controls at every dose of anti-IgM stimulation measured. Thus, IRF4 and c-Rel expression correlate. A certain threshold of c-Rel seems

required for optimal induction of IRF4 in B cells. Indeed, c-Rel dominates the NF-kB program of B cells after antigen-mediated activation[46], potentiating an activated clone for several rounds of proliferation and enabling access to genes associated with terminal differentiation into plasma cells.

Vaccination with antigen in alum, whether used as a prime or a boost, led to an expansion of antigen-specific clones primarily within the GC compartment, generating protracted serum antibody responses that were less protective at early times after immunization compared to the EFR dominated responses generated via antigen plus TLR agonist boosting. This suggests that increasing antigen valency[47] and/ or amounts[48] alone have a limited capacity to direct B cells towards early plasmablast responses following vaccinations, in contrast to vaccines adjuvanted with TLR agonists over an extended period of time. As the data suggest, this may be due to an overall increase in anti-

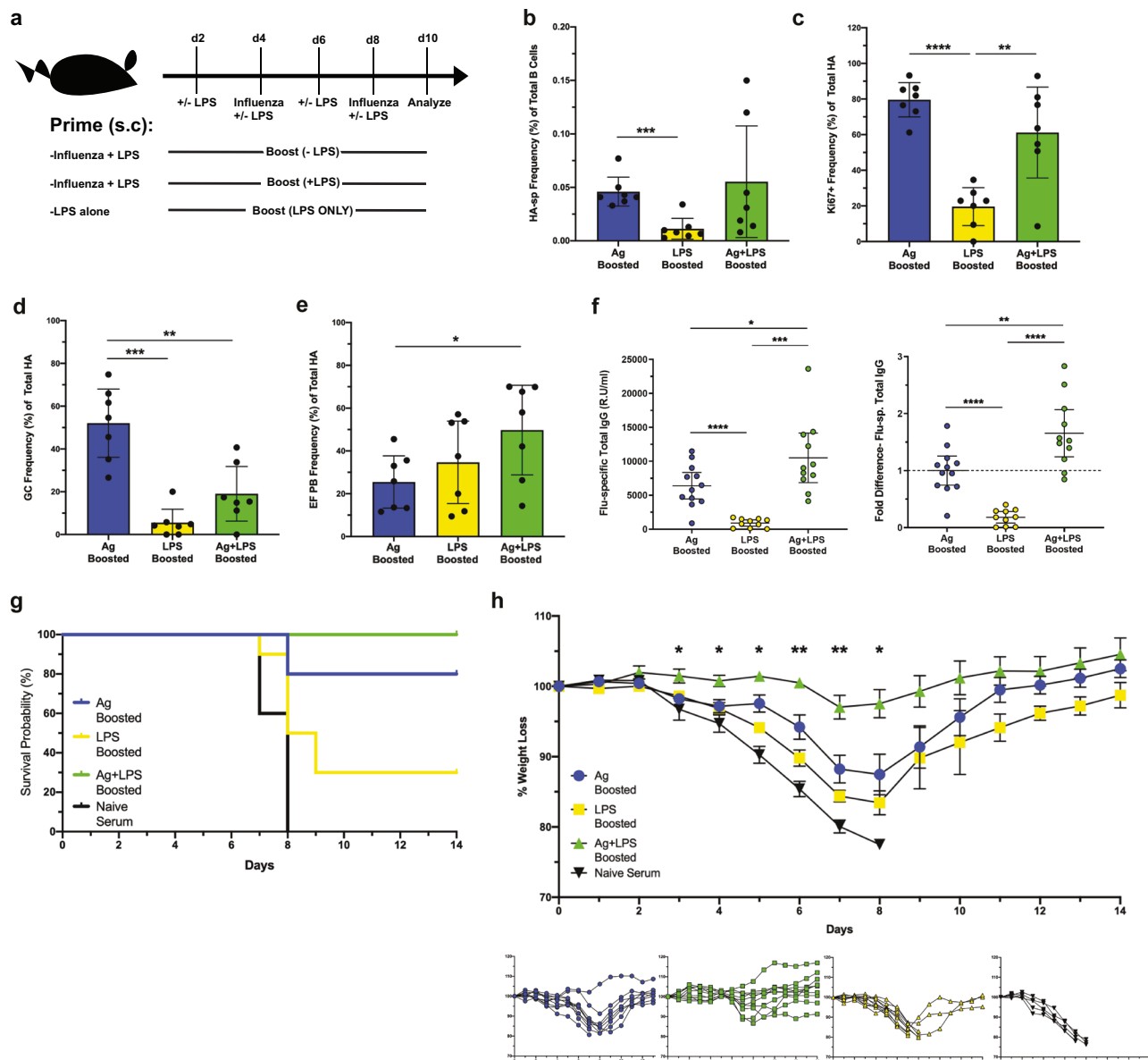

**Fig. 8 | Repeated antigen exposure alone biases antigen-specific B cells towards a GC fate, requires sustained LPS exposure to polarize towards an EF fate.**
**a** C57BL/6 mice (n = 7) were immunized s.c. with influenza and LPS in alum, then boosted with antigen alone or antigen with LPS and LPS alone on days specified, followed by analysis of draining LN. **b**–**e** Quantification of total HA B cells (**b**), Ki67+ HA B cells (**c**), HA GC B cells (**d**) and HA EF PBs (**e**). **f** Concentration of influenza-specific serum IgG at 10 days post-prime (n = 10–12) expressed in relative units (left) and fold difference relative to Antigen Only group (right). **g**, **h** Serum from

primed/boosted mice at 10 days post-prime (n = 10) and naïve mice (n = 6) was transferred to naïve C57BL/6 mice prior to infection with a lethal dose (100 PFU) of influenza A/PR8 the next day. Shown is survival probability (**g**) and percent change in weight (**h**) by average (top) and individually (bottom) over the course of infection. Data in (**b**–**f**, **h**) represent mean ± 95% CI of two (**b**–**e**, **g**, **h**) or three (**d**) independent experiments. Statistical significance determined by one-way ANOVA and two-tailed Student's t-test with Welch's correction. *p < 0.05, **p < 0.01 ***p < 0.001, ****p < 0.0001. Source data are provided as a Source Data file.

influenza antibodies with functional protective capacity under continuous TLR-activating conditions (Fig. 7f) but could also be due to differences in the antibody repertoire that targets unique or more epitopes. As TLR activation provides B cells with increased IRF4 expression, this may allow clones with relatively weaker BCR interactions to partake in antibody secretion by reaching the required IRF4 threshold. Thus, even when the host carries a highly restricted BCR repertoire, TLR activation enables B cells expressing a low affinity BCR to contribute towards immune protection, when otherwise these B cell clones would not reach the threshold of activation and differentiation.

Whether the type of TLR agonist, i.e., one which activates MyD88 or TRIF exclusively, would differentially affect EFR dynamics is unclear. Others showed that the use of TLR4 (MyD88 and TRIF) and TLR7

(MyD88 only) agonists as adjuvant made no difference in early antibody responses after immunization compared to either alone[15]. In addition, TLR9 activation by CpG enhanced titers but worsened the quality of antigen-specific antibody responses due to a lack of GC-mediated affinity maturation to a hapten[19]. Affinity for hapten increases over time as GCs mature and affinity maturation takes place[54], indicating that anti-hapten clones with low avidity for antigen may be 'pulled' into differentiation through activation of TLRs, thus lowering the overall avidity of the response.

Given the PAMPs present in influenza virus, characterization of TLR/BCR synergy upon virus recognition and uptake by B cells, and how this may contribute towards the quality of the antibody response is of interest. It is worth noting that increases in serum antibody

affinities over time were not observed following infection with vesicular stomatitis virus[8,9] and high affinity, germline-encoded antibodies to hemagglutinin were induced early after influenza inoculation[4]. Thus, the level of EFR-derived antibody avidity is contextual and relies on both the inherent specificities of the host's pre-infection repertoire and the valency of epitopes on a given antigen, while the initiation, kinetics, and magnitude of the EFR rely on TLR-mediated inflammatory signals. The data are consistent with findings that memory B cells upon reactivation preferentially form EFR rather than enter GCs, even during heterotypic responses[55]. Given the predominance of TLR-mediated inflammatory signals during acute infection, this allows for antigen-specific B cells to be shunted into EFR for rapid production of protective antibodies to infections. The data also provide a mechanistic explanation for the association of EFRs with severe COVID-19 infection[56], and increased EFR-derived auto-antibody production with chronic inflammation, where a positive feed-forward loop may induce antibody-mediated pathology, driving enhanced inflammation, and thus further supporting ongoing EFRs. As for vaccination, targeting and activation of EFRs would provide for faster production of protective antibodies, whether from naïve B cells, or perhaps even more quickly through memory B cells, allowing for transient protection to take place as the GC response develops. We conclude that B cell response fates are critically regulated by the innate, inflammatory milieu during antigen encounter.

## Methods

### Mice
Male and female 8- to 12-wk-old C57BL/6 (WT; CD45.2 #000664), B6.SJL-Ptprca Pepcb/BoyJ (CD45.1, #002014), B cell–deficient (μMT) mice (#002288), as well as TNFAR1/2 KO (#005540), IFN-gamma KO (#002287), IL-12R KO (#003248), IL-1R KO (#028398), TLR3 KO (#005217), TLR4 KO (#029015), TLR7 KO (#008380) were commercially obtained (The Jackson Laboratories). Breeding pairs of MyD88/TRIF DKO and TLR2/4/unc93b TKO mouse strains were gifts from Dr. Barton (UC Berkeley). Breeding pairs of S100A9 KO mice were a kind gift of Dr. Rafatellu (UC San Diego). Breeding pairs of CD19-Cre IFNAR KO were from Dr. Jason Cyster (UCSF). All mice were housed in SPF housing in ventilated filtertop cages with food and water ad libitum. Euthanasia was done by exposuring mice to $CO_2$. All studies involving mice were conducted in strict compliance with and after approval of protocols by the UC Davis Institutional Animal Care and Use Committee.

Mixed bone marrow (BM) chimeras were generated by adoptively transferring $5 \times 10^6$ total mixed BM cells from sIgM-deficient (CD45.2, 75%) and either C57BL/6 (WT; CD45.2), MyD88/TRIF double knockout (CD45.2), or TLR4/TLR2/Unc93b triple knockout (CD45.2) BM (25%) into 5–6 week-old B6.SJL-Ptprca Pepcb/BoyJ (CD45.1) mice, lethally irradiated by exposure to a gamma irradiation source 24 h prior to transfer. Chimeras were rested for at least 6 weeks before infection and analysis.

### Infections and immunizations
Mice were anesthetized with isoflurane and infected intranasally with a sublethal dose (10 PFU/ml) of influenza A/Puerto Rico/8/34 (A/PR8) in 40 μl volumes in PBS. Virus was grown in hen eggs as previously outlined[57] and each virus batch was titrated for its effect on mice prior to use. Specifically, sublethal infection doses were chosen that incurred no more than 20% weight loss. For immunizations, mice were inoculated subcutaneously with $1 \times 10^7$ PFU A/PR8 in a 50:50 alum to PBS mixture. For some experiments immunizations were supplemented with 3 μg LPS, or mice were in addition boosted repeatedly with $1 \times 10^6$ PFU A/PR8 and 3 μg LPS in PBS or PBS alone as indicated.

### Adoptive serum transfer for passive protection
Indicated strains of mice were infected with 10 PFU A/PR8. Blood from terminally anesthetized mice at 10 dpi was collected via cardiac puncture and spun down for serum separation. Serum from each strain was pooled and naïve C57BL/6 mice were subsequently injected i.v. with a mixture of 50 μl pooled serum and 150 μl 1× PBS. These mice were then inoculated i.n. with 100 PFU A/PR8 one day later and measured for weight loss.

### Magnetic B cell enrichment
Splenic B cells were treated with Fc Block (anti-mouse CD16/32, clone 2.4.G2) and were then enriched using a mixture of biotinylated Abs (anti-CD90.2 (30-H12), anti-CD4 (GK1.5), anti-CD8a (53-6.7), anti-Gr-1 (RB6-8C5), anti-CD11b (M1/70), anti-NK1.1 (PK136), anti-F4/80 (BM8), anti-CD5 (53-7.3), anti-CD9 (MZ3), anti-CD138 (281-2) and anti-biotin MicroBeads (Miltenyi Biotec). Nylon-filtered stained splenocytes were separated using autoMACS (Miltenyi Biotec). Purities of enriched mouse B cells were >98% as determined by subsequent FACS analysis.

### Flow cytometry and phospho-flow
Single-cell suspensions from mediastinal lymph nodes (medLN) were made and labeled for phenotyping as previously outlined[57]. Briefly, after Fc receptor block with anti-CD16/32 (5 mg/ml for 20 min on ice) and Live/dead Fixable Aqua (Thermo Fisher, L34957), cells were stained with the following antibody-fluorophore conjugates at temperatures and times according to manufacturer/provider. All reagents were titrated prior to use to identify dilutions that gave the highest differential fluorescence intensity between the negative and positive cell fraction using mouse spleen, bone marrow or peritoneal cavity wash out cells, as appropriate. Dilutions varied between reagents and reagent lots, but typically fell between 1:25 and 1:400. Higher concentrated reagents (mostly those made in-house) were kept prediluted at a concentration that allowed a 1:200 dilution at use. HA-PE and HA-APC oligomers (kindly provided by Dr. Frances Lund, UAB), BV786 anti-CD19 (1D3) (BD Bioscience, 563333), APC-eFluor780 anti-CD45R (RA3-6B2) (Thermo Fisher, 47-0452-82), PE-Dazzle 594 anti-CD38 (90) (Thermo Fisher, 741748), BV711 anti-CD24 (M1/69) (BD Bioscience, 563450), BV605 anti-CD138 (281-2) (BD Bioscience, 563147), eFluor450 anti-GL-7 (GL7) (Thermo Fisher, 48-5902-82), PE or PE/Cy7 anti-IRF4 (3E4) (Thermo Fisher, 12-9858-82, 25-9858-82), PerCP-eFluor710 anti-IRF8 (V3GYWCH) (Thermo Fisher, 46-9852-82), eFluor450 anti-Ki67 (SolA15) (Thermo Fisher, 48-5698-82), FITC anti-IgM (331) (in-house), and BV650 anti-IgD (11-26c.2a) (Biolegend, 405721). For a non-B cell "dump", the following antibodies on AlexaFluor 700 were used: anti-CD90.2 (Thy1.2) (Biolegend, 105320), anti-CD4 (GK1.5) (Thermo Fisher, 56-0041-82), anti-CD8a (53-6.7) (Thermo Fisher, 56-0081-82), anti-Gr-1 (Thermo Fisher, 56-5931-82), anti-CD11b (M1/70) (Thermo Fisher, 56-0112-82), anti-NK1.1 (Thermo Fisher, 56-5941-82), anti-F4/80 (BM8) (Thermo Fisher, 56-4801-82). The Foxp3 Staining Buffer Set (Thermo Fisher) was used for fixation and permeablization of cells for staining of transcription factors according to the manufacturer's protocol. For cytoplasmic only staining, Cytofix/cytoperm buffer set (BD Biosciences) was used according to the manufacturer's protocol. For phospho-flow, APC anti-p-Syk (moch1ct) (Thermo Fisher, 17-9014-41), PerCP-eFluor710 anti-p-p38 (4NIT4KK) (Thermo Fisher, 17-9078-42), PE/Cy7 anti-p-mTOR (MRRBY) (Thermo Fisher, 25-9718-41), and PE anti-p-p65 (B33B4WP) (Thermo Fisher, 46-9863-42) were stained according to manufacturer's protocol. B cells from 7 dpi medLN were sorted by flow cytometry for ELISPOT using pooled antibodies for dump channel, anti-CD19, anti-CD45R, anti-CD24, and anti-CD38. Purity of sorted cells was assessed immediately afterwards (>96%). Data were collected on BD LSR II Fortessa, BD LSR Symphony, and BD FACS Aria cytometers with BD FACSDiva software, then subsequently analyzed using FlowJo v10 software.

### In vitro B cell cultures
Magnetically enriched B cells were cultured at $5 \times 10^6$ cells/ml at 37 °C. Cells were incubated with anti-IgM (Fab)$_2$ and/or LPS in culture media

at the indicated concentrations for 30 min, one, two, and three hours. Three-hour anti-IgM-pulsed B cells were washed twice with PBS, and then cultured in culture media containing 200 ng/ml CD40L (Pepro-tech) and 5 ng/ml BAFF (R&D Systems) in 96-well round-bottom plates for 48 h at 5% $CO_2$. Subsequent flow cytometric analysis was done using Fc block, Live/dead Fixable Aqua, PE anti-c-Rel (1RELAH5) (Thermo Fisher, 12-6111-80), BV786 anti-CD19, eFluor450 anti-Ki67, PE/Cy7 anti-IRF4, PerCP-eFluor710 anti-IRF8.

### ELISPOT

A/PR8-specific Ig-secreting cells were measured. Briefly, ELISPOT plates were coated with 500 HAU of purified A/PR8 overnight, then blocked for non-specific binding for 1 h. Serial dilutions of FACS-sorted EF PBs and pooled non-EF B cells were incubated overnight at 37 C. Ab-secreting cells (ASC) were revealed with goat anti-mouse IgM, IgG-biotin (South-ern Biotech) followed by SA-HRP (Vector Laboratories) and 3-amino-9-ethylcarbazole (Sigma-Aldrich). ELISPOT images were collected using an AID Elispot Reader and quantified using AID Elispot 7.0.

### Nuclear fraction ELISA

c-Rel nuclear localization was measured. Briefly, nuclear and cyto-plasmic protein fractions were extracted from cultured, purified B cells using NE-PER Nuclear and Cytoplasmic Extraction (Thermo Fisher) according to manufacturer's protocol. ELISA plates were coated at 4 µg/ml dilution of polyclonal anti-c-Rel (Thermo Fisher) overnight, then blocked for non-specific binding for 1 h. Bound c-Rel was detected using 4 µg/ml monoclonal anti-c-Rel (1RELAH5). Binding was revealed by SA-HRP (Vector Laboratories). Data were collected using Molecular Devices SpectraMax M5 and quantified using Softmax Pro 7.

### Viral-load rtPCR

Infected mice were euthanized and lung tissue was extracted and homogenized using Gentle Macs (Miltenyi) in 1 ml PBS. Tissue was pel-leted and supernatant was aliquoted and frozen. Viral RNA was purified from aliquots using the QIAamp viral RNA mini-kit (Qiagen). Presence of influenza was detected through amplification of influenza M gene using rtPCR. Primers used were AM-151 (5′-CATGCAATGGCTAAAGACAA GACC-3′) and AM-397 (5′-AAGTGCACCAGCAGAATAACTGAG-3′) and primer/probe AM-245 (6FAM-5′-CTGCAGCGTAGAGCTTTGTCCAAAA TG-3′-TAMRA). Reverse transcription and amplification were done using TaqPath Multiplex Master Mix (Thermo Fisher). Samples were quanti-fied to a standard of A/PR8 virus stock. Data were collected with Applied Biosystems QuantStudio 6 Flex and quantified using QuantStudio Realtime PCR.

### Calcium flux assay

To measure changes in cellular calcium concentrations, B cells were stained with 2 µM cell-permeant Fluor-3 and 4 µM FuraRed (both Thermo Fisher) according to manufacturer's protocol and stimulated with 10 µg/ml anti-IgM(fab)$_2$ fragments prior to analysis by flow cyto-metry. The ratio of the calcium-excitable (Fluor3) and calcium-quenched (FuraRed) dyes were calculated to determine free-intracellular concentrations.

### Statistics and reproducibility

All statistical analyses were performed with GraphPad Prism v8. Comparisons between two groups were done using a two-tailed Stu-dent's $t$-test with Welch's correction. For more than two groups, a one-way ANOVA was performed followed by a two-tailed Student's $t$-test with Welch's correction between each group. Time courses were ana-lyzed using a two-way ANOVA, followed for each timepoint a two-tailed Students $t$-test with Welch's correction comparing two groups. A $p < 0.05$ was considered statistically significant.

The exact sample size (n) for each experimental group/condition has been provided in each figure legend. Sample size was determined by an initial experiment of $n = 2–4$ in each group (e.g. WT vs KO) then repeated to ensure phenotypic differences were reproducible and statistically significant. No data were excluded from analyses. Most experiments conducted were replicated at least twice. Experiments involving c-Rel nuclear localization were additionally replicated using two different readouts (flow cytometry and ELISA). Certain B cell cul-ture experiments (Suppl. Fig. 5) were partial replicates of explicitly replicated experiments (Fig. 5) and therefore not repeated further. Randomization was not relevant to this study as covariates (e.g. sex, age of animals, cell concentrations in culture) were explicitly matched between experimental and control groups. Blinding was not relevant to this study, as analyses comparing different strains or treatments were done using unilateral cut-offs for each experiment (e.g. flow cytometry gates) or had explicit values generated from a machine/computer (cell counts, ELISA ODs).

### Reporting summary

Further information on research design is available in the Nature Portfolio Reporting Summary linked to this article.

## Data availability

All data supporting the findings of this study are available within the paper, its Supplementary Information, and source data file. Any addi-tional information is available upon request. Source data are provided with this paper.

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

## Acknowledgements

This work was supported by research grants from the NIH/NIAID, R01AI117890, R01AI085568 and U19AI109962 and an institutional NIH training grant from the NIH/NHLBI, T-32 HL007013. We thank Ms. Zheng Luo and Jacqueline Dieter for expert technical support, Drs. Gregory Barton (UC Berkeley), Manuela Raffatellu (UC San Diego) and Jason Cyster (UCSF) for mice, and Dr. Frances Lund (UAB) for HA-baits. We further thank Tracy Rourke of the California National Primate Research Center (UC Davis) for technical assistance with flow cytometry and the UC Davis TRACS personnel for animal care and husbandry.

## Author contributions

J.H.L. designed and conducted experiments, analyzed data, and wrote the manuscript. N.B. obtained funding, designed and supervised experiments, performed data analysis, and wrote the manuscript.

## Competing interests

The authors declare no competing interests.
