## [Peer Review File · Nature Communications]

Toll-like receptor mediated inflammation directs B cells towards protective antiviral extrafollicular responsesREVIEWER COMMENTS

Reviewer #1 (Remarks to the Author):

This manuscript by Lam and Baumgarth investigates the signals that control the generation of the EF versus GC B cell response following infection or vaccination. This is a question of high import given that early control of infection can determine the extent of pathology and severity of illness. The authors reveal a regulatory role of TLR mediated signals in determining the fate decision of activated B cells, with a sustained innate signal promoting differentiation along the EF pathway resulting in early production of antibody capable of promoting clearance. The study includes a careful timecourse of EF versus GC cell and the source of virus-specific antibody. This is an important study that significantly extends our understanding of B cell differentiation. Overall the conclusions are well supported although at times conclusions are based on results that do not appear to be significant (detailed below).

Major comments:

1. One would expect an increase in HA-specific B cells in Fig 3e given the substantial increase in GC cells in the vaccinated animals, but this seems not to be the case. Could the authors comment on the specificity of these new GC cells?
2. What percentage of cells fall into the positive quadrant in non vaccinated mice (Fig. 3d)? As there is no increase in HA-sp B cells from d3 to d7, without a control, can one be certain these are in fact HA specific cells being analyzed?
3. The data in SF2 appear to be total and not HA-specific. Without that, it is difficult to draw conclusions about the HA response.
4. In SF3, there is a trend towards increased EF in TLR7KO mice. The low number of animals and variability in the TLR7 KO mice make achieving significance challenging. If the two types of EF cells are pooled for analysis, does the picture become clearer?
5. In Fig 4b, did the DKO serum provide any protection? Was a control serum transfer experiment performed?
6. The data in SF5 are challenging for comparison between conditions as there are limited times when the same dose is used in combination versus individual stimulations. Nonetheless, one can compare the 1ug/ml anti-IgM and 1ug/ml LPS in this way. When doing so it is hard to see the how the combination strongly supported viability (SF5a) or proliferation (SF5b) compared to the LPS alone as is stated. They appear similar.
7. Are the reductions in Ki67+ cells in SF7 significant? If not this limits the conclusion that can be drawn in lines 247-49.
8. In lines 264-65 authors should state trend as it does not appear to significantly increase. For many of the animals there is a very modest increase and one TKO has a major decrease. Thus conclusions should be drawn with care.
9. Additional discussion of the TLR ligand independent requirement for TLR pathway in B cell activation would be welcome.

Minor comments

- 1.Line 81: believe meant vesicular not vaccinia
- 2.Line 127: add virus after influenza
- 3.Line 197: EFR-derived serum is a bit unclear, suggest "serum from d10 animals wherein antibody is predominantly EFR derived" if this is what is meant
- 4.Line 211: the chimera to which the authors are referring should be noted prior to BMC
- 5.Line 370: induce for indue
- 6.The authors are asked to increase axis labels and percentage values in flow plot font size where possible as many are quite hard to read.
- 7.Consider keeping nomenclature similar- "fold difference" axis label Fig 6e

Reviewer #2 (Remarks to the Author):

The paper by Lam et al explores the mechanisms underlying the formation of extrafollicular plasmablasts (EFRs) and short term humoral memory toward influenza. The authors provide support for a correlation between the formation of EFRs and protective properties of serum at early time points after antigen challenge. Using a series of KO models the authors determine that a key component for the formation of EFRs is Toll like receptor signaling that they find stimulate IRF4 transcription in the activated B-cell. They conclude that inflammatory signals impact the fate of activated B-cells to generate efficient short term response to antigen.

This is in many regards an interesting report representing an impressive amount of work in different model systems. The large number of models and experimental setups explored does, however, become a liability as parts of the data are of limited quality. The authors base many of their conclusions on data collected from few animals creating a challenge in interpretation of the results. This problem is aggravated by that substantial portions of the data are presented as relative controls, in some cases wt, and other cases as compared to non-stimulated cells. It is difficult to fully delineate how this normalization was made. Furthermore, the authors does at times ignore significant differences in their data when they and put forward differences that are not indicated as significant. This creates a challenge to read and validate the data in a proper manner making it difficult to fully appreciate the reported findings and to understand the relevance of the findings.

Specific comments.

- 1: The authors should go over their data and decide what parts that are conclusive and where there is a need to repeat the experiments, alternatively take some data out of the paper or rephrase their conclusions. This is a general problem throughout the report and my list below in mainly to exemplify the problem rather than to provide a complete list of concerns.
- Figure 1, no indication of # of animals or experiments.
 - Figure 1c-e, no significances indicated, hence not possible to support the authors conclusion on row 148 that the response peak at day 9.
 - Figure 2 c-e, no stats indicated. There would appear to be as much HA+GC as HA+EF cells at day 6.
 - The only stats indicated in figure 3b are infection day 10 and 3 days postimmunization. This is not an optimal or relevant comparison. Why have the authors used day 7 data to analyze EF and 10 for GC?
 - In S3a, significant increase in GC is not mentioned. Many data points so spread out that phenotypes well can be lost.
 - In s3b, is there really not any significant difference in EFs in the TLR7 KO?
 - In figure 4b, the TKO display a significant reduction in CD138+EFs that is ignored.
 - In row 204 the authors claim that there is no significant difference between wt and Tko serum in figure 4c. However, the graph shows clear differences day 7 and 8.
 - Row 246 the authors claim that fewer DK and TK cells express KI67, however, no statistical analysis has been performed in figure S7.
 - Row 261, the authors claim that pMtor is increased in TLR signaling deficient cells. However, stat analysis only shown for 0 and 100 Ig concentrations within each one of the genotypes. Hence, I cannot see proper validation of KO vs wt.
 - Row 264, the authors claim that TLR signaling led to enhanced surface Igd expression, however, figure 9a does not show any statistical significances.
 - Row 331 the authors claim that Ag+LPS boosted mice had the highest anti influenza serum levels but in panel 8f, there is no significant difference between Ag boosted and Ag LPS boosted animals.

Minor comments:

- Figure legend for fig 2 contains ref to panels I, j, k not in the figure.
- Row 162-165 talks about smaller GCs. Should this be fewer cells as the size of the GC hardly is investigated?
- 3b GC cells compared to infection day 10 and EF cells day 7.
- Row 214 talks about larger ERFs, correct?
- Are the significance values in figure S8c really correct?

Reviewer #3 (Remarks to the Author):

In this manuscript the authors show that Influenza infection induces rapid early antibody response that is driven by signals from the toll-like receptors (TLRs). This response appears to be protective as seen by serum transfer experiments and inclusion of TLR signals also increases early antibody response during immunization with Influenza antigens. This study is interesting and important in terms of understanding signals regulating early antibody response against viruses and will be important in thinking about vaccine design for Influenza virus. However, the manuscript is presented in manner that makes it difficult understand the relevance and contribution of extrafollicular response. It is known that TLR signals can affect both early B cell responses and germinal center (GC) B cell responses. The main interesting point of this study is the specific effects of TLR signals on extra follicular (EF) response during infection and immunization. But the lack of clarity on what is considered EF response, whether EF response is protective or pathogenic makes it difficult to understand the specific role of EF response. Similarly, it is also not clear which effects of TLR signals are specific to EF B cells and not GC B cells. Details on EF response as well as additional clarification of the data and figures is necessary to appreciate the importance of EF response during Influenza infection and for publication of the manuscript.

Main comments:

It is not described anywhere in the manuscript what the authors consider to be extra follicular response and whether it is the timing or response, location or the markers on the cells that define this type of response. Similarly, many of the figures are based on using CD24 to delineate both GC and EF B cells and again it is not stated anywhere why this strategy was chosen as CD24 is not a marker normally used to delineate GC cells. More details are needed with regards to these.

Strategy for gating of extrafollicular B cells, extrafollicular plasma blasts and germinal center B cells should be explained at least in the first figure on the figure legends and the results section. Similarly in Figure 2 gating strategy for gating on Influenza HA – specific B cells is unclear and there is very little details given in the text or figure legends. Figure 2 legend has multiple errors including mislabeling of the figure legends.

GC B cell data on Figure 1 and 2 or at least just one of the figures should be confirmed with other GC markers apart from CD24 such as PNA or FAS or GL7.

Influenza infection induces early antibody responses in the lungs and recent studies have shown the importance of antigen localization and B cell responses in the lungs (Allie SR et. al 2019 Nature Immunology, Oh EJ Science Immunology 2021) and part of this could also be driven by EF response. It is not clear why the authors decided to investigate only EF response in the mediastinal lymph nodes and not in the lungs which could be more relevant for intranasal infection. Authors could include findings from the lungs or discuss the rationale for only looking at the lymph nodes.

In Figure 4a, the data show that loss of TLR signals lead to specific changes in EF cells but the bone marrow chimera data in Figure 5 b shows that the DKO and TKO chimeras show reduction in both EF and GC responses indicating B cell specific TLR signals have effects on both populations. Similarly, the BCR are responses data are from total B cells showing that changes in TLR signals alters the response of all B cells. Could the authors clarify based on these data how they are concluding that TLR signals specifically effect EF cells? How does the changes in BCR dynamics in all B cells in the knockout condition only lead to effects only on EF cells?

The authors conclude that repeated LPS stimulation polarizes the cells to EF fate, but the data are from time points where you see mostly EF cells and not GC cells, therefore effect is only seen on EF cells. Can the authors look at later time points when GC response are optimal and show that repeated LPS does not have any effect on GC cells?

Could the authors discuss how repeated LPS immunization leads to protective response, is it due to changes in amount of antibody, affinity of antibody or cytokines that might be present in the serum?

Overall, the exact contribution of the EF response during infection is not clear. If EF responses are protective, they should induce viral clearance at early time points? Therefore, in supplementary Fig4a should the viral titre not be higher in the chimeric knockout mice also since they are not able to induce optimal EFR response? If EFR response does not affect viral clearance could the authors discuss how it could be leading to protection ? Similarly, In page 9 it is stated "Thus B cell-intrinsic TLR signaling supports early EFR formation, while additional B cell extrinsic signal further drives EFR generation in a manner that correlates with pathogen burden. How is early EFR response protective and later EFR response pathogenic? Could the authors clarify this and discuss this further as this would be important in thinking of vaccine design?

The data on repeated LPS immunization inducing EFR response are very interesting. Do the authors think this is specific to LPS or inclusion of other TLR ligands such as TLR9 or TLR7 ligands also lead to this feature? The Influenza virus incorporates ssRNA therefore could this be due to stimulation of both TLR4 and TLR7? Additional discussion on this would be useful.

Point-by-point response to reviewer's comments

We thank the reviewers and editors for the thoughtful and constructive comments provided to our manuscript. As we outline below, we have considered and addressed each of the comments provided. This included removal of some data from the supplemental data that lacked statistical power, the strengthening of key findings with additional experimental data and the provision of additional data that demonstrate the induction of protective extrafollicular-derived antibodies against lethal influenza challenge in wild type mice but not those deficient in TLR signaling. Furthermore, we have altered a number of figures and the accompanying text to enhance clarity as requested by the reviewers. We feel that these changes have significantly strengthened the manuscript on both the technical and conceptual aspects of the data. Responses to comments are outlined in blue.

Reviewer #1 (Remarks to the Author):

This manuscript by Lam and Baumgarth investigates the signals that control the generation of the EF versus GC B cell response following infection or vaccination. This is a question of high import given that early control of infection can determine the extent of pathology and severity of illness. The authors reveal a regulatory role of TLR mediated signals in determining the fate decision of activated B cells, with a sustained innate signal promoting differentiation along the EF pathway resulting in early production of antibody capable of promoting clearance. The study includes a careful timecourse of EF versus GC cell and the source of virus-specific antibody. This is an important study that significantly extends our understanding of B cell differentiation. Overall the conclusions are well supported although at times conclusions are based on results that do not appear to be significant (detailed below).

Major comments:

1. One would expect an increase in HA-specific B cells in Fig 3e given the substantial increase in GC cells in the vaccinated animals, but this seems not to be the case. Could the authors comment on the specificity of these new GC cells?

We thank the reviewer for this question. In response we have reanalyzed the data that now show a significant increase in HA-specific B cells between day 0 and day 3, with further increases on day 10 post infection (Fig. 3c-d). Furthermore, we now also demonstrate that HA - binding B cells are enriched for a GC phenotype over an EF phenotype (Fig 3e). The number of HA-specific cells captured with our HA-baits is likely smaller than the total number of HA-specific cells present in the population, as not all HA-specific cells can retain binding to the HA-protein. This is a well known limitation of antigen-specific B cell staining for non-transgenic B cells. In addition, as HA is one of 10 influenza proteins encoded by the virus, other GC cells are directed especially against the nuclear protein and the neuraminidase.

2. What percentage of cells fall into the positive quadrant in non vaccinated mice (Fig. 3d)? As there is no increase in HA-sp B cells from d3 to d7, without a control, can one be certain these are in fact HA specific cells being analyzed?

We thank the reviewer for this question. In response we have made changes to the HA data in Fig. 3c-e, as also outlined above. We are now showing representative flow plots of HA staining on B cells (c), quantifying total HA B cells per timepoint (d), and showing the phenotype (EF vs

GC) of HA B cells per time point (e) post-immunization. We hope this clarifies the identity of the cells in question.

3. The data in SF2 appear to be total and not HA-specific. Without that, it is difficult to draw conclusions about the HA response.

We thank the reviewer for this comment. SF2 does not attempt to specifically address antigen-specific responses. Rather, it supplements Fig. 3's immunization data by showing how EFRs do not respond to greater antigen levels alone, while overall GC sizes do increase upon increased antigen exposure. To clarify and simplify this message, we have incorporated SF2 into subfigures of Fig. 3 as Fig. 3b, which immediately follows the quantification of GC and EF responses in the immunization of original dosage. Although not identifying antigen-specific cells the data very clearly demonstrate the antigen-dose dependent nature of the GC but not the EF response, further supporting our claims that antigen alone does not drive EF responses.

4. In SF3, there is a trend towards increased EF in TLR7KO mice. The low number of animals and variability in the TLR7 KO mice make achieving significance challenging. If the two types of EF cells are pooled for analysis, does the picture become clearer?

This question by the reviewer is much appreciated. We did find statistical significance for TLR7 KO EFRs, but the displayed p-value was lost during editing, we apologize for the oversight. The CD138+ EFs (which are NOT significant) are a subset of the total EF PBs, not a separate population, and therefore are already included in the EF PB population. Text has been added to improve clarification for former SF3 (now SF2) on these populations at Line 188: However, infection of mice lacking TLR3, TLR4, or TLR7 did not result in significant decreases in total EF PBs, nor EF PBs that were CD138+, compared to their WT controls (**Suppl. Fig. 3b**). In fact, there was a slight but significant increase in EFRs of TLR7 KOs. Thus, individual cytokines or innate signaling receptors appeared either not necessary or redundant for EFR development and, in the case of TLR7, may even contribute towards negative regulation.

5. In Fig 4b, did the DKO serum provide any protection? Was a control serum transfer experiment performed?

We thank the reviewer for this question. Control experiments from naïve animal serum transfer of all strains (WT, DKO, TKO) have been added to the revised Fig. 4b as per the reviewer's suggestion. The data demonstrate that serum from naïve mice, independent of their genotype, could not provide immune protection against a lethal influenza virus challenge.

6. The data in SF5 are challenging for comparison between conditions as there are limited times when the same dose is used in combination versus individual stimulations. Nonetheless, one can compare the 1ug/ml anti-IgM and 1ug/ml LPS in this way. When doing so it is hard to see how the combination strongly supported viability (SF5a) or proliferation (SF5b) compared to the LPS alone as is stated. They appear similar.

We thank the reviewer for this comment. In response we have removed some of the data in SF5 and now only compare the most relevant groups, as suggested by the reviewer. The data now clearly demonstrate that 1) TLR signaling has a significantly stronger effect on B cell viability than BCR stimulation at the given doses. However, there is still a small and significant additive effect using both and 2) both TLR and BCR signaling have sizable, additive effects on proliferation and IRF4 upregulation. Clarification has also been added to the relevant text.

7. Are the reductions in Ki67+ cells in SF7 significant? If not this limits the conclusion that can be drawn in lines 247-49.

We thank the reviewer for this comment. After consideration of the importance of demonstrating a difference in Ki67+ cell frequencies, we have decided to remove this particular supplemental data and the associated text. While we are confident that additional sample size would yield significant differences, due to time constraints we opted to omit their repeat. Furthermore, we like to point out that we already demonstrated a relation between TLR signaling and increased proliferation in the in vitro B cell activation data and LPS-boosted immunization data, where we measured Ki67 among HA-specific B cells, providing models that are B cell intrinsic and both intrinsic/extrinsic.

8. In lines 264-65 authors should state trend as it does not appear to significantly increase. For many of the animals there is a very modest increase and one TKO has a major decrease. Thus conclusions should be drawn with care.

We thank the reviewer for this comment have decided to remove the supplemental figure in question. While we are confident that additional sample size would yield significant differences in IgD expression in vivo and support the significant differences found in vitro, it is an observation that is not central to understanding the mechanisms of EF fate decisions explored in this manuscript.

9. Additional discussion of the TLR ligand independent requirement for TLR pathway in B cell activation would be welcome.

We thank the reviewer for the suggestion. In response we have provided additional discussion on the topic of TLR-ligand-independent requirement of optimal BCR activation, stating on line 370 of the amended text: "Evidence of enhanced TLR9-MyD88-BCR complexing in activated B cell-like lymphoma cells suggests that TLRs may provide a platform for downstream TLR targets to become activated through the BCR and its effector pathway (Phelan et al., 2018)".

Consequently, this would make activation of the integrated TLR pathway as the most crucial source of BCR-mediated IRF4."

Minor comments

1.Line 81: believe meant vesicular not vaccinia

This error has been corrected.

2.Line 127: add virus after influenza

This error has been corrected.

3.Line 197: EFR-derived serum is a bit unclear, suggest "serum from d10 animals wherein antibody is predominantly EFR derived" if this is what is meant

This edit has been made, we thank the reviewer for this suggestion.

4.Line 211: the chimera to which the authors are referring should be noted prior to BMC

This error has been corrected.

5.Line 370: induce for indue

The error has been corrected.

6. The authors are asked to increase axis labels and percentage values in flow plot font size where possible as many are quite hard to read.

Axes and labels have been increased where needed, we thank the reviewer for this suggestion.

7. Consider keeping nomenclature similar- “fold difference” axis label Fig 6e

The axis label has been changed, we thank the reviewer for this suggestion.

Reviewer #2 (Remarks to the Author):

The paper by Lam et al explores the mechanisms underlying the formation of extrafollicular plasmablasts (EFRs) and short term humoral memory toward influenza. The authors provide support for a correlation between the formation of EFRs and protective properties of serum at early time points after antigen challenge. Using a series of KO models the authors determine that a key component for the formation of EFRs is Toll like receptor signaling that they find stimulate IRF4 transcription in the activated B-cell. They conclude that inflammatory signals impact the fate of activated B-cells to generate efficient short term response to antigen.

This is in many regards an interesting report representing an impressive amount of work in different model systems. The large number of models and experimental setups explored does, however, become a liability as parts of the data are of limited quality. The authors base many of their conclusions on data collected from few animals creating a challenge in interpretation of the results. This problem is aggravated by that substantial portions of the data are presented as relative controls, in some cases wt, and other cases as compared to non-stimulated cells. It is difficult to fully delineate how this normalization was made. Furthermore, the authors does at times ignore significant differences in their data when they and put forward differences that are not indicated as significant. This creates a challenge to read and validate the data in a proper manner making it difficult to fully appreciate the reported findings and to understand the relevance of the findings.

Specific comments.

1. The authors should go over their data and decide what parts that are conclusive and where there is a need to repeat the experiments, alternatively take some data out of the paper or rephrase their conclusions. This is a general problem throughout the report and my list below in mainly to exemplify the problem rather than to provide a complete list of concerns.

We thank the reviewer for these suggestions and have taken specific steps to clarify data representation and conclusions drawn, some in response also to comments from reviewer #1. We now explicitly describe fold-differences to WT controls in figure legends and text for appropriate data. Data lacking significance have been removed (chimera HA-specific Ki67 and IgD expression) or repeated (immunization with LPS boosting experiments in Figs. 7 and 8). The results have strengthened and further confirmed our initial conclusions.

2. Figure 1, no indication of # of animals or experiments.

We apologize for this oversight. The number of animals and experiments conducted have now been added to the legend for Figure 1.

3. Figure 1c-e, no significances indicated, hence not possible to support the authors conclusion on row 148 that the response peak at day 9.

We thank the reviewer for this comment and have conducted statistical analysis using one-way ANOVAs on each time course measurement (Fig. 1c-e). Significance has now been indicated at the amended Fig. 1 and the corresponding text has been clarified.

4. Figure 2 c-e, no stats indicated. There would appear to be as much HA+GC as HA+EF cells at day 6.

One-way ANOVAs were conducted on each time course measurement (Fig. 2c-e) and significance has now been indicated in the amended Figure. The observation by the reviewer that HA+GC and HA+EF cell frequencies are similar was likely based on the fact that we had used y-axes of different scales to indicate small changes in the GC populations. As the most important comparison here is between GC and EF responses and to avoid that potential confusion, we have now adjusted both y-axes for Figs. 2d and 2e to be the same.

5. The only stats indicated in figure 3b are infection day 10 and 3 days postimmunization. This is not an optimal or relevant comparison. Why have the authors used day 7 data to analyze EF and 10 for GC?

We thank the reviewer for this comment. We had included the data simply to indicate the large differences between these responses, but agree with the reviewer that this is hard to justify. Therefore, we have made several changes to Figure 3 to better articulate that GC responses are the dominant B cell fate during immunization and that EF responses are minimal. As the reviewer has suggested, we have removed the infection timepoint comparisons and focused solely on characterizing the immunization response. Fig. 3a now quantifies EF and GC responses, while data from SF2 has been incorporated as Fig. 3b to show that increasing antigen dose does not increase or rescue total EF responses, while it does increase overall GC responses in an antigen dose-dependent manner. Furthermore, Figs. 3c-e have been reconfigured to better show EF vs GC fate of HA-specific B cells, showing their expansion after immunization in Fig. 3c - d, and characterizing their phenotype (EF vs GC) in Fig. 3e. We believe this amended figure more clearly demonstrates that immunization s.c. favors GC responses and provides contrast to primary infection's early bias towards EF responses.

6. In S3a, significant increase in GC is not mentioned. Many data points so spread out that phenotypes well can be lost.

We thank the reviewer for this comment. We have now added text to the manuscript to mention the significant increase in GC responses in mice lacking TNF α signaling at line 183.

We acknowledge the reviewer's concern that some data points are spread out in some of the many groups of gene-targeted mice we screened (we only show the most pertinent ones we tested in the paper), we are confident that each of the genotypes tested failed to show measurable reductions in EFRs at the chosen timepoint, thus that these genes/signaling pathways are no critical for early EFR formation after influenza virus infection. The data were always obtained by simultaneously testing the various KO mice against age and sex-matched congenic strains of C57BL/6 mice and for numerous of these comparisons we performed ELISA for virus-specific serum antibodies. Because these ELISA data failed to show any significant differences, and because day 7 is a rather early timepoint to measure antibody levels, we are not showing this additional data in the manuscript.

7. In s3b, is there really not any significant difference in EFs in the TLR7 KO?

We apologize for the omission, also remarked by reviewer #1. Indeed, we did find statistical significance for TLR7 KO EFRs, but the displayed p-value was lost during editing, we apologize for the oversight. The CD138+ EFs (which are NOT significant) are a subset of the total EF PBs, not a separate population, and therefore are already included in the EF PB population. Text has been added to improve clarification for former SF3 (now SF2) on these populations at Line 188: However, infection of mice lacking TLR3, TLR4, or TLR7 did not result in significant decreases in total EF PBs, nor EF PBs that were CD138+, compared to their WT controls (**Suppl. Fig. 3b**). In fact, there was a slight but significant increase in EFRs of TLR7 KOs. Thus, individual cytokines or innate signaling receptors appeared either not necessary or redundant for EFR development and, in the case of TLR7, may even contribute towards negative regulation.

8. In figure 4b, the TKO display a significant reduction in CD138+EFs that is ignored.

We thank the reviewer for this comment. In response we now added to the amended manuscript that there was a significant reduction in TKO CD138+ EFs (Fig. 4a) at line 206: Surprisingly, infection of another TLR-null model, through deletion of genes for TLR2³⁷, TLR4³⁸ and a missense mutation of Unc93b³⁹ (TKO), showed EFRs similar to WT controls (**Fig. 4a**) along with nominal passive protective capacity (**Fig. 4c**), despite slight reductions in CD138+ EF PBs at 7 dpi (**Fig. 4a**).

9. In row 204 the authors claim that there is no significant difference between wt and Tko serum in figure 4c. However, the graph shows clear differences day 7 and 8.

We thank the reviewer for this comment and have now changed the wording of the results section for Fig. 4c to specifically address the outcome of TKO serum transfer in these protection experiments on Line 205: “Surprisingly, infection of another TLR-null model, through deletion of genes for TLR2³⁷, TLR4³⁸ and a missense mutation of Unc93b³⁹ (TKO), showed EFRs similar to WT controls (**Fig. 4a**) along with nominal passive protective capacity (**Fig. 4c**), despite slight reductions in CD138+ EF PBs at 7 dpi (**Fig. 4a**).”

10. Row 246 the authors claim that fewer DK and TK cells express KI67, however, no statistical analysis has been performed in figure S7.

We thank the reviewer for this comment also made by reviewer #1. As outlined above, we decided to remove the supplemental figure and associated text, as we have shown this to be the case in other parts of the manuscript.

11. Row 261, the authors claim that pMtor is increased in TLR signaling deficient cells.

However, stat analysis only shown for 0 and 100 Ig concentrations within each one of the genotypes. Hence, I cannot see proper validation of KO vs wt.

We apologize for this oversight. In the revised manuscript we have now added the statistical analysis comparing each treatment from each KO to their respective WT treatment control, indicated by stars above each individual treatment condition. The associated text has been clarified/changed as well to better reflect the data.

12. Row 264, the authors claim that TLR signaling led to enhanced surface IgD expression, however, figure 9a does not show any statistical significances.

We thank the reviewer for this comment. As also remarked in response to reviewer #1, we have decided to remove the supplemental figure in question. While we are confident that additional sample size would yield significant differences in IgD expression *in vivo* and support the

significant differences found *in vitro*, it is a supplemental observation that does not specifically address the question of the mechanisms of EF fate decisions explored in this manuscript.

13. Row 331 the authors claim that Ag+LPS boosted mice had the highest anti influenza serum levels but in panel 8f, there is no significant difference between Ag boosted and Ag LPS boosted animals.

We thank the reviewer for this comment. In response we have conducted additional experiments and added the data to the figure. The data now clearly demonstrate significant differences between Ag+LPS and Antigen only (Fig. 8f) both when expressed as relative units total flu-specific IgG (left panel) and when showing normalized data to the average "Antigen Only" IgG levels from each experiment, then expressed as fold-change. The data demonstrate the significant increases in antigen-specific, serum IgG when LPS boosting occurs in both absolute and relative terms, explaining the enhanced passive protective capacity of the sera from these mice, shown in Fig. 8g-h.

Minor comments:

1. Figure legend for fig 2 contains ref to panels l, j, k not in the figure.

This error has been corrected, we apologize for the oversight.

2. Row 162-165 talks about smaller GCs. Should this be fewer cells as the size of the GC hardly is investigated?

This error has been corrected.

3. 3b GC cells compared to infection day 10 and EF cells day 7.

We believe we have addressed this above (Major Comment #5) and made the appropriate changes to the figure in question (Fig. 3).

4. Row 214 talks about larger ERFs, correct?

The text has been clarified to address the reviewers comment.

5. Are the significance values in figure S8c really correct?

We thank the reviewer for this question. The significances shown in (former) SF8c are from one-way ANOVAs of each strain to demonstrate that different anti-IgM treatment concentrations were actually causing changes in the measured protein's phosphorylation signature. With the changes made from the reviewer's previous comment on the figure, the effect on p38 in TLR-null B cells has been clarified.

Reviewer #3 (Remarks to the Author):

In this manuscript the authors show that Influenza infection induces rapid early antibody response that is driven by signals from the toll-like receptors (TLRs). This response appears to be protective as seen by serum transfer experiments and inclusion of TLR signals also increases early antibody response during immunization with Influenza antigens. This study is interesting and important in terms of understanding signals regulating early antibody response against viruses and will be important in thinking about vaccine design for Influenza virus. However, the manuscript is presented in manner that makes it difficult understand the relevance and contribution of extrafollicular response. It is known that TLR signals can affect both early B

cell responses and germinal center (GC) B cell responses. The main interesting point of this study is the specific effects of TLR signals on extra follicular (EF) response during infection and immunization. But the lack of clarity on what is considered EF response, whether EF response is protective or pathogenic makes it difficult to understand the specific role of EF response. Similarly, it is also not clear which effects of TLR signals are specific to EF B cells and not GC B cells. Details on EF response as well as additional clarification of the data and figures is necessary to appreciate the importance of EF response during Influenza infection and for publication of the manuscript.

Main comments:

1. It is not described anywhere in the manuscript what the authors consider to be extra follicular response and whether it is the timing or response, location or the markers on the cells that define this type of response. Similarly, many of the figures are based on using CD24 to delineate both GC and EF B cells and again it is not stated anywhere why this strategy was chosen as CD24 is not a marker normally used to delineate GC cells. More details are needed with regards to these.

We thank the reviewer for this comment. In response we have added additional details to the description of EFRs in the introduction of the amended manuscript at Line 59: "Instead, early antibodies are produced from short-lived plasmablasts of the extrafollicular response (EFR), which develop and localize within the medulla and interfollicular regions² of the respiratory tract-draining mediastinal lymph nodes (medLN) shortly after infection and before GC formation³." Furthermore, on Line 67: "EFRs thus appear physiologically distinct from GCs and can generate protective, germline-encoded, antigen-specific ASCs from the restrictive repertoire of inbred mice."

We and others previously demonstrated that CD24 staining can delineate B cell subsets. Specifically, that CD45R+/PNA+ cells identified as GC B cells had high CD24 expression and additionally co-stained as GL7+ and Fas+ (Shinall et al., 2000, JI; Baumgarth, 2004, Methods Cell Biol.; Elsner et al. 2015 PloSPathog). Nonetheless, to more clearly delineate our gating strategy we have updated Fig. 1 with added labels and we show the expression of GL7 and relatively high IRF8 on CD24^{hi} gated GC and expression of CD138 and high IRF4, in EF PB, respectively.

2. Strategy for gating of extrafollicular B cells, extrafollicular plasma blasts and germinal center B cells should be explained at least in the first figure on the figure legends and the results section. Similarly in Figure 2 gating strategy for gating on Influenza HA – specific B cells is unclear and there is very little details given in the text or figure legends. Figure 2 legend has multiple errors including mislabeling of the figure legends.

We thank the reviewer for this comment. In response we have made multiple changes to the relevant text in the results section and figure legends to Figs 1 and 2 to clarify gating strategy. Furthermore, labels were added to Fig. 1 to aid the identification of target cell populations. We hope that these changes are clarifying identification of each cell subset.

3. GC B cell data on Figure 1 and 2 or at least just one of the figures should be confirmed with other GC markers apart from CD24 such as PNA or FAS or GL7.

We thank the reviewer for this suggestion. CD24/CD38 gating for GCs is now confirmed by GL7 staining in Figure 1 along with IRF8 expression level as outlined in response to the previous comment. In addition, we refer to our previous publications that identify this gating strategy as

appropriate as these cells are PNA+ FAS+ and GL7+ (Elsner et al, 2015 PloSPathog), the latter now added to the Figure.

4. Influenza infection induces early antibody responses in the lungs and recent studies have shown the importance of antigen localization and B cell responses in the lungs (Allie SR et. al 2019 Nature Immunology, Oh EJ Science Immunology 2021) and part of this could also be driven by EF response. It is not clear why the authors decided to investigate only EF response in the mediastinal lymph nodes and not in the lungs which could be more relevant for intranasal infection. Authors could include findings from the lungs or discuss the rationale for only looking at the lymph nodes.

We thank the reviewer for this comment. In response, we have now added to the text at the beginning of the results section that “Influenza-specific ASCs can be found predominantly in the medLN within 7 days after primary infection but are not found in the lungs until 14 dpi³, well after virus clearance. This indicated that medLN EFRs are the main source of the early antigen-specific antibody response and were thus investigated.”

5. In Figure 4a, the data show that loss of TLR signals lead to specific changes in EF cells but the bone marrow chimera data in Figure 5 b shows that the DKO and TKO chimeras show reduction in both EF and GC responses indicating B cell specific TLR signals have effects on both populations. Similarly, the BCR are responses data are from total B cells showing that changes in TLR signals alters the response of all B cells. Could the authors clarify based on these data how they are concluding that TLR signals specifically effect EF cells? How does the changes in BCR dynamics in all B cells in the knockout condition only lead to effects only on EF cells?

We thank the reviewer for this comment and acknowledge the observations made about differences in phenotypes between the knockout models, with the TLR-null bone-marrow chimera data and *in vitro* data showing a universal effect on B cell activation, suggesting an effect on both EF and GC, while the global TLR knockout *in vivo* data demonstrated a more specific effect on EF responses. Firstly, the global knockout and chimera infection data demonstrate that there is a bona-fide B cell-intrinsic effect through TLRs on the EF response, not that it is exclusive to the EF response. We have ensured the text reflects this conclusion carefully as well. However, this does indicate that the GC response can be rescued if non-B cells also lack TLR signaling. As the focus of this paper is on the EF response, we did not explore the mechanisms by which GC responses are rescued in a global TLR-null context, but have provided a possible explanation in the discussion at Line 361: “As TLR signaling in DCs leads to increased Th1 polarization⁵⁰, perhaps a total ablation of TLR signaling may polarize more CD4 T cells towards a Tfh phenotype, compensating for the GC B cell-intrinsic defect in TLR-mediated activation”.

6. The authors conclude that repeated LPS stimulation polarizes the cells to EF fate, but the data are from time points where you see mostly EF cells and not GC cells, therefore effect is only seen on EF cells. Can the authors look at later time points when GC response are optimal and show that repeated LPS does not have any effect on GC cells?

We thank the reviewer for this question. Additional data has been added (**Suppl. Fig. 8**) that shows GCs were not affected by the LPS-boosting regimen. Thus, demonstrating no detrimental effect on the development of GC responses. Text addressing these data have been added as well.

7. Could the authors discuss how repeated LPS immunization leads to protective response, is it due to changes in amount of antibody, affinity of antibody or cytokines that might be present in the serum?

Additional discussion on how repeated LPS boosting leads to more protection has been included as per the reviewer's request at Line 398: "As the data suggest, this may be due to an overall increase in anti-influenza antibodies with functional protective capacity under continuous TLR-activating conditions (**Fig. 7f**) but could also be due to differences in antibody quality, i.e. differences in repertoire that target unique or a higher number of epitopes. As TLR activation provides B cells with increased IRF4 expression, this allows for clones with relatively weaker BCR interactions to partake in antibody secretion by reaching the required IRF4 threshold."

8. Overall, the exact contribution of the EF response during infection is not clear. If EF responses are protective, they should induce viral clearance at early time points? Therefore, in supplementary Fig4a should the viral titre not be higher in the chimeric knockout mice also since they are not able to induce optimal EFR response? If EFR response does not affect viral clearance could the authors discuss how it could be leading to protection? Similarly, in page 9 it is stated "Thus B cell-intrinsic TLR signaling supports early EFR formation, while additional B cell extrinsic signal further drives EFR generation in a manner that correlates with pathogen burden. How is early EFR response protective and later EFR response pathogenic? Could the authors clarify this and discuss this further as this would be important in thinking of vaccine design?"

We thank the reviewer for these pertinent questions. The follow text has been added to address the reviewer's questions on the contributions of EFRs towards virus clearance at Line 70: "Addressing how these distinct B cell activation outcomes contribute to humoral immunity against acute respiratory tract virus infections, where rapid induction of immunity is a key determinant of survival, is pertinent for our understanding of the pathogenesis of these infections and the role of B cell immunity. While CD8 T cells are credited the most in clearance of influenza during primary infection, they alone cannot prevent mortality⁵ and may collaterally target non-infected antigen-presenting cells⁶. Additionally, lack of B cells lead to a ~50 fold increase in virus titers by day 10 post-infection⁷ demonstrating the potential importance of early antibody generation against influenza. This has important implications for vaccine design, as vaccines are generally considered only successful if inducing GC-derived, long-lived plasma cells and memory B cells. However, vaccinations during the ongoing COVID-19 pandemic or during seasonal influenza virus infections are likely more effective if they can induce immune protection more quickly, i. e. through EFRs."

As for addressing if EFRs are protective vs pathogenic, we demonstrate here that EFRs during influenza are protective when MyD88/TRIF signaling is replete and that this protective effect is lost when MyD88/TRIF is absent, but upstream TLR ablation does not lead to this defect. Differences in virus clearance between global TLR-null mice and B cell-specific, TLR-null chimeras demonstrate the contribution of TLR signaling in both manners, with TLR-null chimeras having replete TLR signaling in non-B cells that allows for optimal innate and T cell responses leading to nominal virus clearance (**Suppl. Fig. 3a**), despite disruptions in both B cell responses (**Fig. 5b**).

9. The data on repeated LPS immunization inducing EFR response are very interesting. Do the authors think this is specific to LPS or inclusion of other TLR ligands such as TLR9 or TLR7

ligands also lead to this feature? The Influenza virus incorporates ssRNA therefore could this be due to stimulation of both TLR4 and TLR7? Additional discussion on this would be useful.

We thank the reviewer for this comment and have addressed questions regarding specific TLR activation with the following text at Line 408: "Whether the type of TLR agonist, i.e. one which activates MyD88 or TRIF exclusively, would differentially affect EFR dynamics is unclear. As stated previously, using TLR4 (MyD88 and TRIF) and TLR7 (MyD88 only) adjuvant made no difference in early antibody responses after immunization compared to either alone¹⁵.

Additionally, TLR9 activation by CpG enhanced titers but worsened the quality of antigen-specific antibody responses due to a lack of GC-mediated affinity maturation to the hallmark antigen hapten¹⁹. Affinity for hapten increases over time as GCs mature and affinity maturation takes place⁵⁴, indicating that clones with low avidity interactions with hapten may be 'pulled' into differentiation through activation of TLRs, thus lowering the overall avidity of the response.

Given the PAMPs present in influenza virus, characterization of TLR/BCR synergy upon virus recognition and uptake by B cells, and how this may contribute towards plasmablast formation, would be of interest. Yet increases in serum antibody affinities over time were not observed following infection with vesicular stomatitis virus^{8,9} and high affinity, germline-encoded antibodies to hemagglutinin were induced early after influenza inoculation⁴."

REVIEWERS' COMMENTS

Reviewer #1 (Remarks to the Author):

The authors have addressed the concerns raised.

I note a few edits that are needed.

1. Line 333 and "=" mark
2. Line 399 "in in"
3. Line 407 Was the or meant to be an and?

Reviewer #2 (Remarks to the Author):

The paper is now of acceptable quality and the general story holds up. I do, however, believe that the authors should be more careful in future submissions as it is not reasonable that reviewers should act as proofreaders of submitted reports.

Reviewer #3 (Remarks to the Author):

The authors have addressed all of my concerns from the previous review , including adding new data related to markers for GC cells and clarification of phenotype and function of extra follicular cells. Additional comments in the discussion about the role of TLR signaling are also adequate.

Point-by-point response to reviewer's comments:

We'd like to thank all the reviewers for their comments that improved this manuscript's message and data robustness/fidelity. It is much appreciated.

Reviewer #1 (Remarks to the Author):

The authors have addressed the concerns raised.

We thank the reviewer for their helpful feedback in making this a stronger manuscript.

I note a few edits that are needed.

1. Line 333 and "=" mark

The error has been corrected, apologies for the oversight.

2. Line 399 "in in"

This error has been corrected, apologies for the oversight.

3. Line 407 Was the or meant to be an and?

This error has been corrected, apologies for the oversight.

Reviewer #2 (Remarks to the Author):

The paper is now of acceptable quality and the general story holds up. I do, however, believe that the authors should be more careful in future submissions as it is not reasonable that reviewers should act as proofreaders of submitted reports.

We thank the reviewer for their helpful feedback, apologize for the original manuscript appeared insufficiently edited/proofread.

Reviewer #3 (Remarks to the Author):

The authors have addressed all of my concerns from the previous review, including adding new data related to markers for GC cells and clarification of phenotype and function of extra follicular cells.

Additional comments in the discussion about the role of TLR signaling are also adequate.

We thank the reviewer for their helpful feedback in making this a stronger manuscript.